# Antioxidant properties of dihydroxy B-ring flavonoids modulate circadian amplitude in *Arabidopsis*

Evan S Littleton[1,2,3] , Sherry B Hildreth[2], Shihoko Kojima[1,2] , Brenda SJ Winkel[1,2]

**Flavonoids are abundant specialized metabolites produced by plants for a range of functions, including pigmentation, hormonal signaling, UV protection, and drought tolerance. We previously showed that flavonoids also influence the circadian clock in *Arabidopsis*. Here, we report that the antioxidant properties of dihydroxy B-ring flavonoids are responsible for influencing the amplitude of the core clock gene luciferase reporter, *TOC1:LUC*. We found the elevated amplitude of *TOC1:LUC* rhythms correlates with the elevated cellular $H_2O_2$ content of flavonoid-deficient seedlings. Moreover, two different chemical approaches to reduce the accumulation of reactive oxygen species rescued the elevated *TOC1:LUC* amplitude in flavonoid-deficient seedlings, whereas chemically reducing auxin transport, a known function of flavonoids, had no impact. Interestingly, $Ca^{2+}$ levels in the chloroplast, but not the cytosol, were also altered in flavonoid-deficient seedlings, hinting at retrograde signaling as a possible target of flavonoid action. This study advances our understanding of the relationship between flavonoids and the circadian clock, as well as the mechanisms underlying this interaction.**

## Introduction

The pervasiveness of circadian rhythms across the tree of life is indicative of the evolutionary benefit provided by the ability to predict and prepare for a changing daily environment (Jabbur & Johnson, 2022; Oravec & Greenham, 2022; Laosuntisuk et al, 2023). In plants, circadian rhythms control nearly all aspects of physiology, including cellular metabolism, photosynthetic activity, growth, stomatal opening, defense from pathogens and herbivores, leaf movement, and flowering (Karapetyan & Dong, 2018; Creux & Harmer, 2019; Venkat & Muneer, 2022). Plants that do not match the pace of their internal clock to the external environment, either through altered circadian period or through arrhythmia, have decreased chlorophyll content, $CO_2$ assimilation, and biomass (Dodd et al, 2005). In addition to the period/pace of the clock,

there is now growing evidence for the importance of amplitude/robustness, as a robust clock has been co-selected with agriculturally important crop traits, particularly flowering time. The recently named field of "chronoculture" proposes to target the plant circadian clock through genetic modification or timing-based agricultural practices to further enhance crop productivity and agricultural sustainability (Creux & Harmer, 2019; Steed et al, 2021).

At the cellular level, circadian rhythms are generated by a transcription–translation feedback loop (TTFL) consisting of multiple complex interactions between transcription factors (Creux & Harmer, 2019; Laosuntisuk et al, 2023). In plants, the evening gene, *TOC1*, and the morning genes, *CCA1/LHY*, are central players in a network of overlapping feedback loops that generates a robust 24-h cycle of transcription and translation regulating thousands of downstream genes (Harmer et al, 2000; Nagel et al, 2015; Liebelt et al, 2019). Although the TTFL is predominantly synchronized by external input from the environment such as light, temperature, and humidity (Millar, 2004; Salomé & McClung, 2005; Mwimba et al, 2018), recent studies have shown the importance of inputs from internal sources as well; clock gene amplitude, period, or phase can be modified through photosynthetic sugars (Dalchau et al, 2011; Haydon et al, 2013, 2017; Frank et al, 2018; Román et al, 2021), cytosolic free calcium (Martí Ruiz et al, 2018), and reactive oxygen species (ROS) (Zhou et al, 2015; Román et al, 2021). These studies showcase the innate sensitivity and adaptability of the plant clock to both environmental cues and internal cellular changes, as well as highlight the importance of reciprocal signaling from metabolites as a critical aspect of how the TTFL is controlled.

We recently showed that a loss of flavonoid biosynthesis in *Arabidopsis* modulates clock gene expression, affecting both transcript levels and the amplitude of *TOC1:LUC* rhythms, without significantly altering phase or period (Hildreth et al, 2022). This influence on clock function adds to a long list of well-established roles of flavonoids in plants, including functions in pigmentation, development, and stress protection. Many of these roles have been attributed to the strong antioxidant potential of flavonoids, particularly the dihydroxy B-ring forms such as quercetin and cyanidin (Fig S1; Agati et al, 2012; Agati et al, 2020; Daryanavard et al, 2023; Gayomba & Muday, 2020; Hernández et al, 2009; Nakabayashi et al,

[1]Department of Biological Sciences, Virginia Tech, Blacksburg, VA, USA [2]Fralin Life Sciences Institute, Virginia Tech, Blacksburg, VA, USA [3]Molecular and Cellular Biology Graduate Program, Virginia Tech, Blacksburg, VA, USA

Correspondence: winkel@vt.edu

2014; Xu et al, 2017). A recent study has uncovered a role of superoxide as a metabolic signal that affects the amplitude of clock gene expression (Román et al, 2021), prompting us to investigate the possibility that a loss of antioxidant activity is responsible for the elevated *TOC1:LUC* amplitude in flavonoid mutants. Here, we show that a deficiency of dihydroxy B-ring flavonoids in *Arabidopsis* seedlings enhances the amplitude of *TOC1:LUC*, predominantly via the elevated ROS level. Using biochemical and genetic approaches, we begin to elucidate the mechanisms underlying the ROS-dependent modulation of clock gene expression by flavonoids.

# Results

### Exogenously supplied flavonoids affect *TOC1* promoter activity

In a previous study, we found that bioluminescence output from *TOC1:LUC* had a higher amplitude in *tt4-11* seedlings relative to WT controls (Hildreth et al, 2020). The *tt4-11* line lacks all flavonoids because of a T-DNA insertion in the gene encoding chalcone synthase (CHS), the first enzyme in the flavonoid biosynthetic pathway (Bowerman et al, 2012; Fig S1). Bioluminescence output from *TOC1:LUC* also had a higher amplitude in *tt7-5* mutants, which lack the enzyme, flavonoid 3'-hydroxylase (F3'H), also because of a T-DNA insertion (Bowerman et al, 2012), and can therefore produce monohydroxy but not dihydroxy B-ring flavonoids. Furthermore, we showed that although the amplitude of the *CCA1:LUC* reporter is not altered in these lines, supplementing the medium with the dihydroxylated flavonol, quercetin, slightly enhanced *CCA1:LUC* amplitude in both WT and *tt4-11* seedlings. These data led us to hypothesize that dihydroxy B-ring flavonoids play a predominant role in modulating circadian amplitude.

To test this hypothesis, we first focused on the impact of exogenously applied flavonoids on the expression of the *TOC1:LUC*, as this reporter exhibits enhanced amplitude in flavonoid mutant lines, *tt4-11* and *tt7-5* (Hildreth et al, 2022). Seeds for the Col-0 WT, *tt4-11*, and *tt7-5* lines containing the *TOC1:LUC* reporter were sown on medium containing naringenin, the first intermediate in flavonoid biosynthesis (Fig S1). After 6-d incubation in a 12-h light: 12-h dark (LD) cycle, the seedlings were transferred to constant darkness (DD) for luminescence measurements. For both Col-0 and *tt4-11*, the amplitude of *TOC1:LUC* bioluminescence was slightly enhanced in the presence of 1 μM naringenin compared with the control (DMSO), whereas 10 μM naringenin had no effect (Figs 1A and B and S2A; Supplemental Data 1). At 100 μM, however, *TOC1:LUC* amplitude decreased in Col-0 and decreased even more drastically in *tt4-11*. No changes were observed in the period of *TOC1:LUC* luminescence with these treatments (Fig S3A), consistent with our previous report (Hildreth et al, 2022). In contrast, treatment with 100 μM naringenin did not have any effect on *TOC1:LUC* amplitude in *tt7-5*, which cannot convert naringenin to dihydroxy B-ring flavonoids (Fig 1C and D). These data support our hypothesis that the dihydroxy B-ring forms of flavonoids are primarily responsible for the changes in the amplitude of *TOC1:LUC* rhythms.

We next tested the effects of the predominant flavonoid types produced in *Arabidopsis* on *TOC1:LUC* expression. We found that at 100 μM, kaempferol and quercetin (monohydroxy and dihydroxy B-ring flavonols, respectively) both lowered *TOC1:LUC* amplitude in *tt4-11*, whereas cyanidin (a dihydroxy B-ring anthocyanidin) fully reduced amplitude to WT control levels (Figs 1E and S2B). In contrast to our observations for *tt4-11*, exogenous flavonoids had only a minor effect on *TOC1:LUC* amplitude in Col-0, although cyanidin did appear to consistently reduce reporter amplitude relative to WT controls. Although the similar effects of the two flavonols were unexpected, this may be the result of conversion of kaempferol to quercetin through the action of F3'H (e.g., Ueyama et al, 2002; Zhou et al, 2016), which remains fully functional in *tt4-11*. It also appears that cyanidin has an even greater impact on *TOC1: LUC* amplitude in *tt4-11* than either flavonol, although differential uptake or conversion/modification cannot be ruled out at this stage. Consistent with the effects observed with naringenin, the period of *TOC1:LUC* expression remained unchanged under all conditions (Fig S3B). Overall, these findings show that exogenous application of different flavonoid subtypes, and particularly cyanidin, a dihydroxy B-ring anthocyanidin, can restore the enhanced amplitude in flavonoid-deficient seedlings back to WT levels.

To determine whether exogenous flavonoid treatments led to accumulation of flavonoids *in planta*, we performed untargeted LC-MS/MS after treatment with 100 μM naringenin in Col-0, *tt4-11*, and *tt7-5*, as well as 100 μM kaempferol, quercetin, or cyanidin in Col-0 and *tt4-11*. As expected, untreated *tt4-11* had no detectable flavonoid glycosides, whereas untreated *tt7-5* accumulated excess kaempferol glycosides but no detectable quercetin or isorhamnetin glycosides (Fig S4), consistent with previous reports (Gayomba & Muday, 2020). Moreover, naringenin treatment of *tt4-11* restored accumulation of kaempferol, quercetin, and isorhamnetin glycosides, whereas in *tt7-5*, this treatment increased accumulation of kaempferol glycosides but not quercetin or isorhamnetin glycosides. However, kaempferol and quercetin treatment had minimal effects on the accumulation of flavonoids, likely because of the limited uptake and transport of flavonols to aboveground tissues (Buer et al, 2007).

### Few flavonol glycosides exhibit rhythmicity across the day–night cycle

The expression of flavonoid biosynthetic genes is highly rhythmic at the mRNA level (Harmer et al, 2000; Nagel et al, 2015; Liebelt et al, 2019); however, little is known about the accumulation patterns of the pathway's end products across the day–night cycle in seedlings. Because there is often a poor correlation between the transcriptome and metabolome even within the same pathway (Hildreth et al, 2020), we asked whether specific flavonoids exhibit rhythmic patterns that could underlie modulation of clock gene amplitude. To this end, we used untargeted LC-MS/MS to examine metabolite profiles over 24 h in 7-d-old Col-0 seedlings grown under a 12-h:12-h LD cycle, where zeitgeber time 0 and 12 represent the onset and offset of light, respectively. Of the eight flavonoid glycosides identified in our dataset (predominantly kaempferol, quercetin, and isorhamnetin glycosides), only the monohydroxylated flavonol, kaempferol deoxyhexose, and the

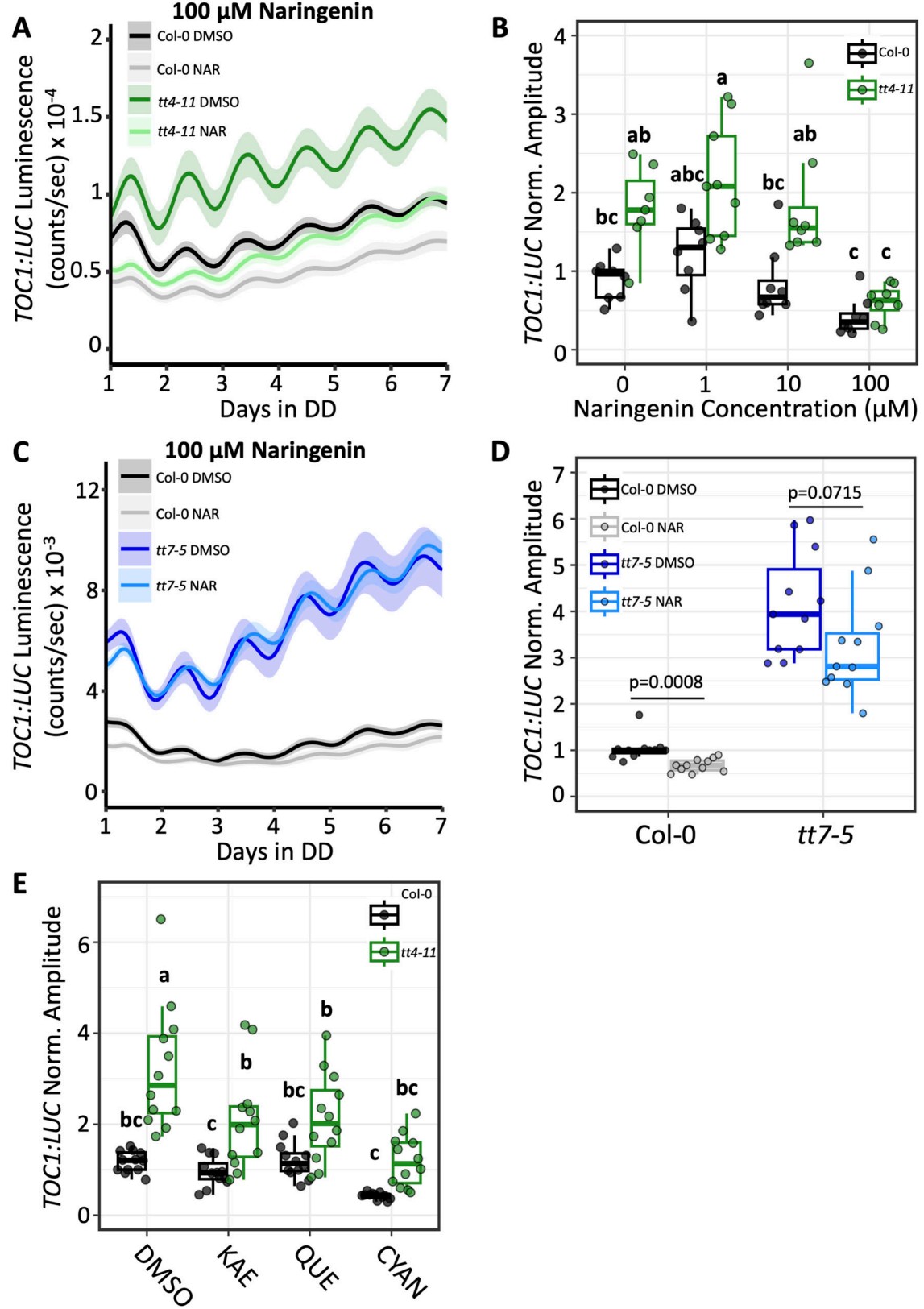

**Figure 1. Effects of exogenous flavonoids on *TOC1:LUC* amplitude in WT and flavonoid-deficient lines.**
**(A)** Bioluminescent output of *TOC1:LUC* in Col-0 or *tt4-11* seedlings grown on 0.1% DMSO (black: Col-0, green: *tt4-11*) or 100 μM naringenin (NAR) (gray: Col-0, light green: *tt4-11*) (n = 8 biological replicates from two independent experiments). **(B)** Amplitude of *TOC1:LUC* (black: Col-0, green: *tt4-11*) with 0, 1, 10, or 100 μM NAR. Values were normalized to Col-0 0 μM naringenin (0.1% DMSO), which was set to 1. **(C)** Bioluminescent output of *TOC1:LUC* in Col-0 or *tt7-5* seedlings grown on 0.1% DMSO (black: Col-0,

dihydroxylated flavonol, isorhamnetin hexose deoxyhexose, showed a rhythmic pattern of accumulation at a statistically significant level. All other flavonoid glycosides showed a weak diurnal pattern, but with very low amplitude and statistically insignificant rhythmicity (Fig 2; Supplemental Data 2). These observations in seedlings are consistent with a recent finding that flavonoid glycosides do not accumulate rhythmically in LD or LL in *Arabidopsis* leaves (Rivière et al, 2024). However, it remains possible that specific flavonoids responsible for modulating clock behavior have stronger rhythmic accumulation patterns that are only discernible at the cellular or organellar level.

### Effects of flavonoids on *TOC1:LUC* amplitude do not appear to involve auxin transport

A well-established function of flavonoids is the inhibition of auxin transport (Brown et al, 2001; Buer & Muday, 2004; Lewis et al, 2011). This is believed to occur through interactions of flavonols such as quercetin and kaempferol with PIN-FORMED (PIN) or ABCB transporters responsible for intracellular movement of auxin (Teale et al, 2021; Daryanavard et al, 2023). Interestingly, previous studies found that several clock gene reporters, including *TOC1: LUC*, exhibited reduced amplitude when seedlings were grown in the presence of indole-3-acetic acid (IAA), the most common form of auxin found in plants (Hanano et al, 2006; Covington & Harmer, 2007). To test the possibility that the increased auxin transport, and resulting altered auxin distribution, in *tt4-11* (Buer & Muday, 2004) causes the elevation in *TOC1:LUC* amplitude, we grew seedlings in the presence of N-1-naphthylphthalamic acid (NPA), which inhibits auxin transport similar to flavonoids by interacting with PINs and ABCB transporters (Teale et al, 2021). When we used a range of concentrations that were previously shown to reduce root gravitropism and growth (Rashotte et al, 2000; Brown et al, 2001), NPA had no significant impact on *TOC1:LUC* amplitude or period in either Col-0 or *tt4-11* seedlings (Figs 3A and B, S2C, and S3C; Supplemental Data 1). Together, these findings suggest that the mechanism by which flavonoids influence *TOC1:LUC* amplitude does not involve their role in inhibiting auxin transport.

### *TOC1:LUC* amplitude correlates with $H_2O_2$ content

Because of the strong antioxidant capabilities of dihydroxy B-ring flavonoids (Agati et al, 2012, 2020; Nakabayashi et al, 2014) and ability of ROS to modulate circadian amplitude (Zhou et al, 2015; Román et al, 2021), we next asked whether the elevated amplitude of *TOC1:LUC* observed in *tt4-11* is driven by elevated ROS levels. To test this, we first measured $H_2O_2$ levels in *tt4-11* and Col-0 seedling lysates across 24-h cycles under different light conditions (Fig 4A). Consistent with previous reports of elevated $H_2O_2$ and $O_2^-$ in

flavonoid mutant seedlings (Xu et al, 2017; Chapman & Muday, 2021), we found that in LD (12-h light/12-h dark), the $H_2O_2$ levels were ~3-fold higher in *tt4-11* than in WT seedlings at all time points (Fig 4B). The levels of $H_2O_2$ also remained two- to threefold higher in *tt4-11* after transfer from LD to constant light (LL) or constant darkness (DD) (Fig 4C and D).

We then asked whether exogenous flavonoids, which influenced *TOC1:LUC* amplitude (Fig 1), also affect $H_2O_2$ levels. The level of $H_2O_2$ in the WT seedlings was unchanged in the presence of NAR, KAE, QUE, or CYAN (Fig 4E). In contrast, the level of $H_2O_2$ in *tt4-11* seedlings was slightly, but insignificantly, reduced by KAE, QUE, or CYAN, whereas it was fully restored to WT levels by NAR. To determine whether the dihydroxy B-ring forms of flavonoids are primarily responsible for modulating $H_2O_2$ levels, we examined the effects of exogenous flavonoids on $H_2O_2$ levels in *tt7-5*. In contrast to the effects in *tt4-11*, NAR only partially reduced the $H_2O_2$ level in the *tt7-5* seedlings relative to WT (Fig 4F). These results indicate that a lack of dihydroxy B-ring flavonoids is primarily responsible for the elevated $H_2O_2$ levels in flavonoid mutant lines.

To better determine whether the antioxidant activity of flavonoids is responsible for suppressing the amplitude of *TOC1:LUC*, we next performed a correlation analysis between *TOC1:LUC* amplitude and $H_2O_2$ levels across all three genotypes and four flavonoid treatment groups. Using a linear regression model, we saw a positive linear correlation between *TOC1:LUC* amplitude and $H_2O_2$ content (Fig 4G), strongly suggesting that flavonoids, and particularly the dihydroxy B-ring forms, influence the amplitude of *TOC1: LUC* via their antioxidant activities.

We then asked whether reducing ROS levels in *tt4-11* would lower the elevated *TOC1:LUC* amplitude in these seedlings. To this end, we first used a chemical approach, treating seedlings with diphenyleneiodonium (DPI). This inhibitor of the NADPH oxidases that generate $O_2^-$, a precursor of $H_2O_2$, at the plasma membrane, has previously been shown to strongly attenuate the enhancement of *TOC1:LUC* amplitude in WT seedlings in response to sucrose (Román et al, 2021). DPI treatment of *tt4-11* led to a significant dose-dependent reduction in *TOC1:LUC* amplitude, with 10 and 30 $\mu$M treatments resulting in amplitudes that were equal to or lower than, respectively, those in untreated WT seedlings (Figs 5A and B and S2D, Supplemental Data 1). DPI also decreased *TOC1:LUC* amplitude in Col-0, although the difference was not statistically significant. Similar to treatment with flavonoids, no changes in the period of *TOC1:LUC* expression were observed at any concentration of DPI (Fig S3D).

We next tested another approach to reducing intracellular ROS, and treated the seedlings with the $H_2O_2$ scavenger, potassium iodide (KI) (Agati et al, 2020; Gayomba & Muday, 2020), in which the iodide ion ($I^-$) reacts with free $H_2O_2$ to generate diatomic iodine ($I_2$) and $H_2O$. At 100 $\mu$M KI, the amplitude of *TOC1:LUC* was reduced in *tt4-11* seedlings relative to untreated controls, whereas there was no

blue: *tt7-5*) or 100 $\mu$M NAR (gray: Col-0, light blue: *tt7-5*) (n = 10–11 biological replicates from three independent experiments). **(D)** Amplitude of *TOC1:LUC* in Col-0 or *tt7-5* grown on 0.1% DMSO or 100 $\mu$M NAR. Values were normalized to Col-0 DMSO, which was set to 1. **(E)** Amplitude of *TOC1:LUC* (black: Col-0, green: *tt4-11*) with 0.1% DMSO or 100 $\mu$M kaempferol (KAE), quercetin (QUE), or cyanidin (CYAN) (n = 12 biological replicates from three independent experiments). Values were normalized to Col-0 DMSO, which was set to 1. *P*-values in (D) are calculated from a two-tailed *t* test (Col-0; unequal variance, *tt7-5*; equal variance). Letters in (B, E) represent grouping from one-way ANOVA followed by Tukey's post hoc test with a significance cutoff of *P* < 0.05. In (B, D, E), boxplot midlines represent the median value (Q2); lower and upper lines represent 25th (Q1) and 75th percentiles (Q3), respectively. Whiskers represent the range of data within 1.5 interquartile range (Q3-Q1) from Q1 or Q3. **(A, C)** Solid line and shading in (A, C) represent mean ± SEM.

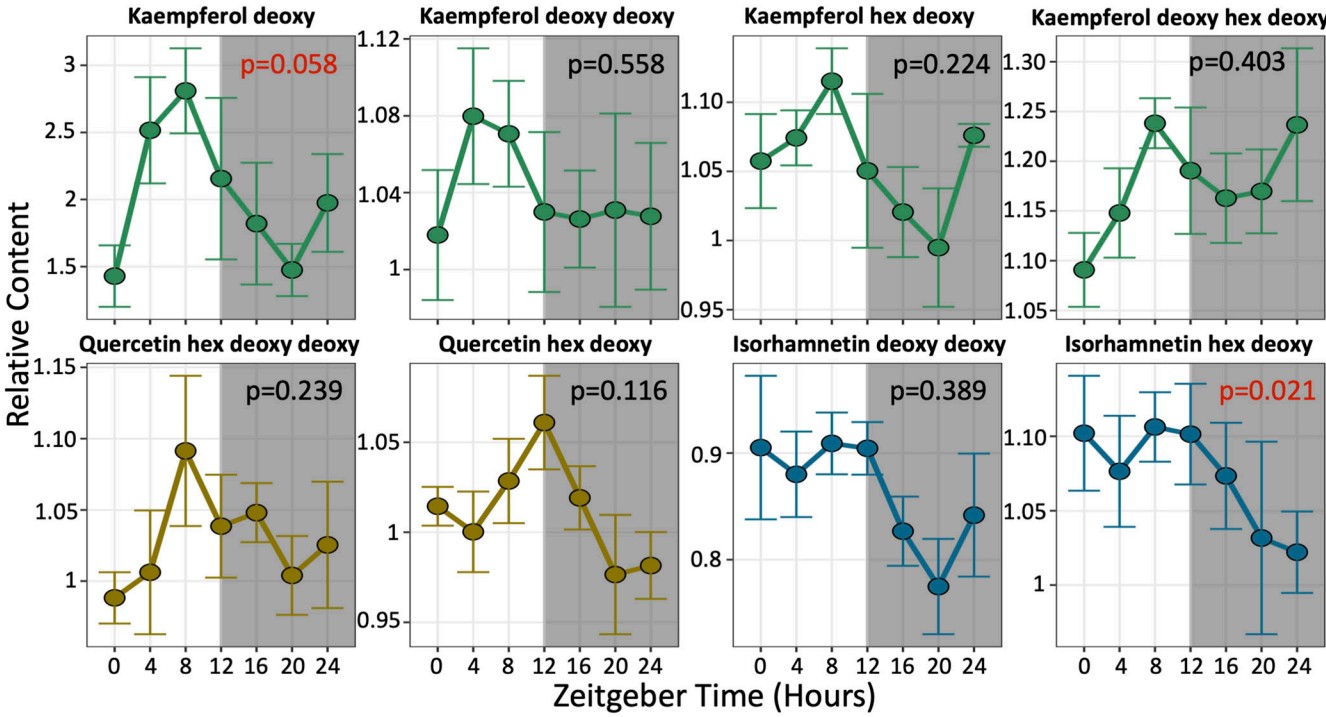

**Figure 2. Flavonoid accumulation patterns in WT seedlings.**
Methanolic extracts from 7-d-old Col-0 plants grown under LD were examined by LC-MS (n = 4–5 biological replicates for each time point). All the data represent the mean ± SEM. Peak area values were normalized to replicate #1 from time point 0, which was set to 1. *P*-values for rhythmicity are calculated from MetaCycle with algorithms JTK, LS, and ARS. MetaCycle *P*-values that were considered statistically significant (*P* < 0.1) are highlighted in red. Abbreviations for glycosides: hex, hexose; deoxy, deoxyhexose.

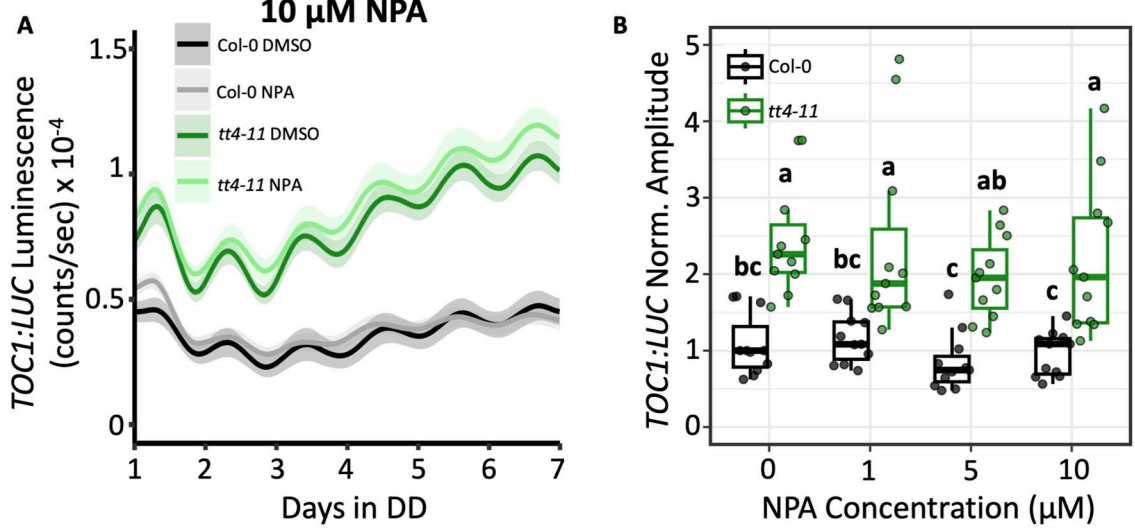

**Figure 3. Effects of inhibition of auxin transport on *TOC1:LUC* amplitude.**
**(A)** Bioluminescent output of *TOC1:LUC* in Col-0 or *tt4-11* seedlings grown on 0.1% DMSO (black: Col-0, green: *tt4-11*) or 10 µM NPA (gray: Col-0, light green: *tt4-11*). Solid line and shading represent the mean ± SEM (n = 11 biological replicates from three independent experiments). **(B)** Amplitude of *TOC1:LUC* (black: Col-0, green: *tt4-11*) grown on 0, 1, 5, or 10 µM NPA. Values were normalized to Col-0 0 µM NPA (0.1% DMSO), which was set to 1. Letters in (B) represent grouping from one-way ANOVA followed by Tukey's post hoc test with a significance cutoff of *P* < 0.05. Boxplot midlines represent the median value (Q2); lower and upper lines represent 25th (Q1) and 75th percentiles (Q3), respectively. Whiskers represent the range of data within 1.5 interquartile range (Q3-Q1) from Q1 or Q3.

impact in Col-0 (Fig 5C and D). Together with the effects observed for treatment with DPI, these results show that the enhanced amplitude of *TOC1:LUC* in *tt4-11* can be lowered by reducing ROS levels either by inhibiting $O_2^-$ generation or by supplementing with $H_2O_2$ scavengers. These findings provide further evidence that flavonoids suppress *TOC1:LUC* amplitude through their antioxidant properties.

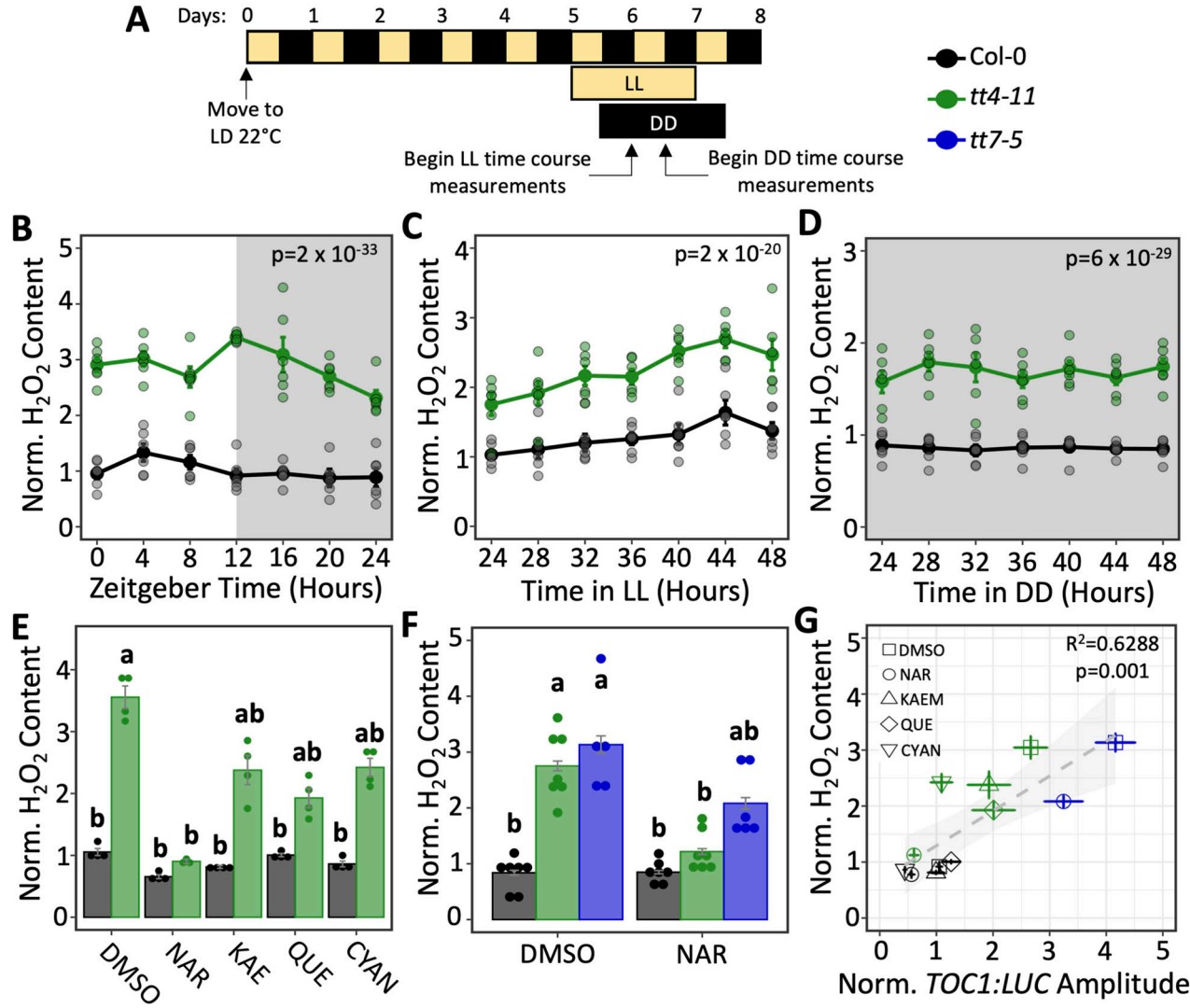

**Figure 4. Correlation between H₂O₂ levels and *TOC1:LUC* amplitude.**
**(A)** Diagram of experimental timeline for sample preparation and collection. **(B)** H₂O₂ content of 6-d-old seedlings (black: Col-0, green: *tt4-11*) collected every 4 h for 24 h in LD (n = 5–6 biological replicates/time point from two independent experiments). Values were normalized to Col-0 zeitgeber time 0, which was set to 1. **(C)** H₂O₂ content of 6-d-old seedlings (black: Col-0, green: *tt4-11*) transferred to LL for 24 h and collected every 4 h for 24 h (n = 6 biological replicates/time point from two independent experiments). Values were normalized to Col-0 time point 24, which was set to 1. **(D)** H₂O₂ content of 6-d-old seedlings (black: Col-0, green: *tt4-11*) transferred to DD for 24 h and collected every 4 h for 24 h (n = 6 biological replicates/time point from two independent experiments). Values were normalized to Col-0 time point 24, which was set to 1. **(E)** H₂O₂ content of 6-d-old seedlings (black: Col-0, green: *tt4-11*) grown on media containing DMSO or various flavonols transferred to DD for 24 h before collection (n = 3–4 biological replicates from two independent experiments). Values were normalized to Col-0 DMSO, which was set to 1. **(F)** H₂O₂ content of 6-d-old seedlings (black: Col-0, green: *tt4-11*, blue: *tt7-5*) grown on media containing DMSO or NAR transferred to DD for 24 h before collection (n = 5–7 biological replicates/time point from three independent experiments). Values were normalized to Col-0 DMSO, which was set to 1. **(G)** Linear regression model between *TOC1:LUC* amplitude and H₂O₂ content. *P*-values in (B, C, D) are calculated from two-way ANOVA comparing the average H₂O₂ content over time between genotypes. Letters in (E, F) represent grouping from one-way ANOVA followed by Tukey's post hoc test with a significance cutoff of *P* < 0.05. All data represent the mean ± SEM.

## Modulation of *TOC1:LUC* amplitude by flavonoids does not involve the NPR1 receptor

We next attempted to gain mechanistic insights into how the antioxidant activity of flavonoids influences the amplitude of *TOC1: LUC*. We first focused on NPR1 (non-expressor of pathogenesis-related gene 1), a master regulator of the plant immune response, and hypothesized that flavonoid modulation of H₂O₂ levels

regulates the core clock through NPR1. This is because NPR1 controls the expression of *TOC1* and other clock genes via the plant's redox state (Zhou et al, 2015), and because naringenin induces nuclear localization of NPR1 in a ROS-dependent manner (An et al, 2021). If our hypothesis was correct, we predicted that the changes in the amplitude of the *TOC1:LUC* reporter by exogenous flavonoids (Fig 1) would be lost in the absence of NPR1. The amplitude of *TOC1:LUC* was lower in *npr1-3*, a NPR1 null line, than Col-0

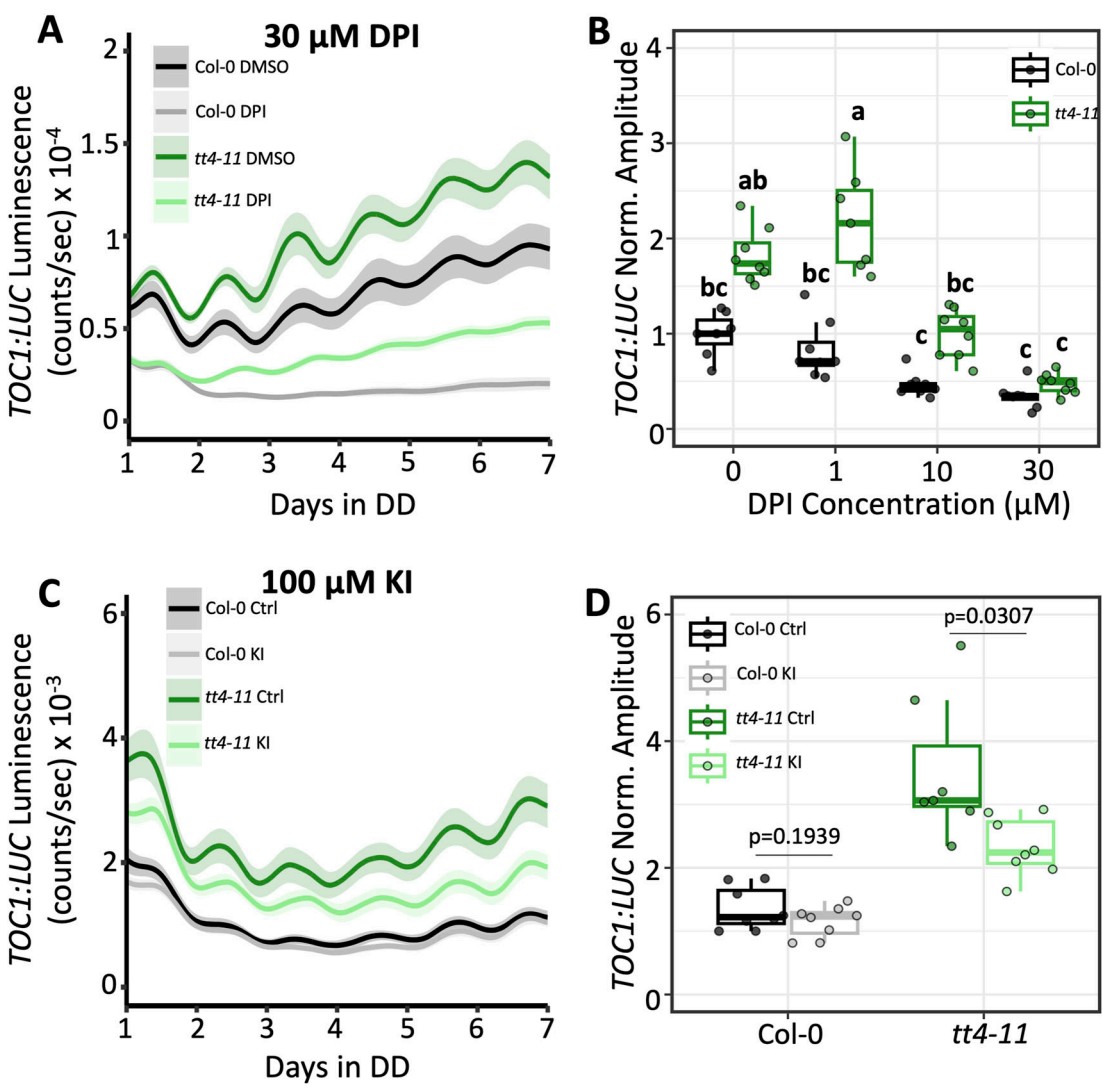

**Figure 5. Effect of chemically-induced reactive oxygen species reduction on *TOC1:LUC* amplitude in *tt4-11*.**
**(A)** Bioluminescent output of *TOC1:LUC* in Col-0 or *tt4-11* seedlings treated with 0.1% DMSO (black: Col-0, green: *tt4-11*) or 30 $\mu$M DPI (gray: Col-0, light green: *tt4-11*) (n = 7–8 biological replicates from two independent experiments). **(B)** Amplitude of *TOC1:LUC* in seedlings (black: Col-0, green: *tt4-11*) treated with 0, 1, 10, or 30 $\mu$M DPI. Values were normalized to Col-0 at 0 $\mu$M DPI (0.1% DMSO), which was set to 1. **(C)** Bioluminescent output of *TOC1:LUC* in seedlings (black: Col-0, green: *tt4-11*) grown on untreated media or 100 $\mu$M KI (gray: Col-0, light green: *tt4-11*) (n = 7–8 biological replicates from two independent experiments). **(D)** Amplitude of *TOC1:LUC* in seedlings (black: Col-0, green: *tt4-11*) treated with 0 (Ctrl) or 100 $\mu$M KI. Values were normalized to Col-0 Control, which was set to 1. *P*-values in (D) are calculated with a two-tailed *t* test with equal variance. Letters in (B) represent grouping from one-way ANOVA followed by Tukey's post hoc test with a significance cutoff of $P < 0.05$. In (B, D), boxplot midlines represent the median value (Q2); lower and upper lines represent 25th (Q1) and 75th percentiles (Q3), respectively. Whiskers represent the range of data within 1.5 interquartile range (Q3-Q1) from Q1 or Q3. **(A, C)** Solid line and shading in (A, C) represent the mean ± SEM.

seedlings, consistent with the previous report (Zhou et al, 2015). However, the amplitude and the basal level of *TOC1:LUC* were reduced in the presence of 100 $\mu$M naringenin relative to untreated controls (Fig S5A and B), in both Col-0 and *npr1-3*. These data indicate that the effect of flavonoids on the amplitude of *TOC1:LUC* is independent of NPR1.

**Flavonoids may influence clock gene expression through Ca²⁺ signaling in chloroplasts**

We next focused on cytosolic free calcium level ($[Ca^{2+}]_{cyt}$) and hypothesized the increased $H_2O_2$ content in *tt4-11* (Fig 4) alters

daily $[Ca^{2+}]_{cyt}$ rhythms, which in turn lead to the elevated amplitude of *TOC1:LUC*. This is because changes to $[Ca^{2+}]_{cyt}$ alter the expression of clock genes through a pathway involving *TOC1* (Martí Ruiz et al, 2018). Moreover, ROS increases $[Ca^{2+}]_{cyt}$ in plants by altering the activity of $Ca^{2+}$ channels/transporters (Mori & Schroeder, 2004; Mazars et al, 2010; Fichman et al, 2022; Ravi et al, 2023). To address this, we used the *Arabidopsis* MAQ2 line expressing cytosolic aequorin, a luminescent $Ca^{2+}$ biosensor (Knight et al, 1991; Johnson et al, 1995), and crossed this with Col plants carrying the *tt4-2* allele (Bennett et al, 2006) to circumvent silencing effects from the T-DNA insertion in *tt4-11* (Daxinger et al, 2008). *tt4-2* also lacks a functional CHS enzyme, in this case

because of a point mutation that disrupts pre-mRNA splicing (Burbulis et al, 1996). When we monitored the bioluminescent output from cytosolic aequorin (MAQ2) using 6-d-old LD-grown seedlings transferred to DD, we saw no changes in the amplitude of circadian $[Ca^{2+}]_{cyt}$ oscillations in *tt4-2* compared with Col-0 (Fig 6A and C).

We also measured the circadian free calcium rhythms in chloroplasts ($[Ca^{2+}]_{chl}$) in Col-0 and *tt4-2* by performing crosses with the *Arabidopsis* MAQ6 line expressing chloroplast-targeted aequorin (Johnson et al, 1995; Lenzoni & Knight, 2018). Interestingly, the luminescence of this reporter indicated that $[Ca^{2+}]_{chl}$ was elevated in *tt4-2* relative to Col-0, although there was no difference in the amplitude of rhythmic $[Ca^{2+}]_{chl}$ (Fig 6B and D). The *tt4-2* seedlings also exhibited a significant reduction in the well-established spike in $[Ca^{2+}]_{chl}$ that occurs at the transition from light to dark (Fig 6B) (Johnson et al, 1995; Wood et al, 2001; Martí Ruiz et al, 2018; Pivato et al, 2023; Kuang et al, 2024 *Preprint*). We also asked whether the same effect on the light–dark $[Ca^{2+}]_{chl}$ spike occurs in *tt7-1*, an F3'H-deficient line in the La-er background containing a point mutation that creates a stop codon in the first exon of the F3'H gene (Schoenbohm et al, 2000). Like *tt4-2*, this mutant line exhibited a significant reduction in the light–dark $[Ca^{2+}]_{chl}$ spike relative to the parental La-er line (Fig S6A). The reduced $[Ca^{2+}]_{chl}$ spike in *tt4-2* and *tt7-1* was not due to lower MAQ6 expression in these lines because the total amount of reconstituted aequorin in individual seedlings showed no significant difference in luminescence between Col-0 and *tt4-2* seedlings expressing chloroplast-targeted aequorin, whereas *tt7-1* seedlings expressing this reporter had even a slightly higher luminescence relative to La-er (Fig S6B and C).

Together, these data suggest that the absence of dihydroxy B-ring flavonoids in *Arabidopsis* influences $Ca^{2+}$ homeostasis in the chloroplast but not the cytosol, and that a loss of antioxidant potential could affect crosstalk between $H_2O_2$ and $Ca^{2+}$ signaling pathways (e.g., Bhattacharyya et al, 2025), specifically in chloroplasts, a major site of ROS production. This finding also points to retrograde signaling between the chloroplast and the nucleus as a potential target of flavonoid action that indirectly affects *TOC1:LUC* amplitude.

## Discussion

We previously reported that the lack of CHS, the first enzyme in the flavonoid pathway, leads to the dysregulation of core clock gene expression in *Arabidopsis* and increased the amplitude of *TOC1: LUC* (Hildreth et al, 2022). These findings indicate that the circadian clock is influenced by flavonoid metabolism, although the underlying mechanism is unknown. In this study, we focused on the change in the amplitude of *TOC1:LUC* to examine this question, hypothesizing that flavonoids, rather than the enzyme itself, are the important mediators of this regulation, and testing which characteristics of flavonoids underlie this effect. We found that antioxidant activity, but not activation of the NPR1 receptor, one of the few identified players in redox modulation of the core clock (Zhou et al, 2015), or altered auxin transport was important for the action of flavonoids on *TOC1:LUC* amplitude (Figs 3, 4, 5, and S5). In addition, although changes in cytoplasmic $[Ca^{2+}]$ were not detected

in plants devoid of flavonoids, the *tt4-2* and *tt7-1* lines did exhibit changes in $Ca^{2+}$ levels in the chloroplast (Figs 6 and S6). Interestingly, our analyses further indicate that the hydroxylation pattern of the B ring is a significant factor influencing redox homeostasis and clock amplitude (Fig 4), as for many other physiological processes in plants and in animals.

Many prior reports have highlighted the potent antioxidant potential of dihydroxy B-ring flavonoids as a prime biochemical property of this class of compounds. In vitro experiments have consistently shown that quercetin, a dihydroxy B-ring flavonol, has a higher antioxidant activity than kaempferol, its mono-hydroxylated counterpart (Pietta, 2000; Agati et al, 2012; Nakabayashi et al, 2014; Csepregi & Hideg, 2018). In vivo studies have also noted that dihydroxy B-ring flavonoids have a significant impact on redox homeostasis (Hernández et al, 2009; Nakabayashi et al, 2014; Xu et al, 2017; Agati et al, 2020; Gayomba & Muday, 2020; Daryanavard et al, 2023). In our study, all of the effects of flavonoids on *TOC1:LUC* amplitude appear to rely on the ability of seedlings to produce dihydroxy B-ring flavonoids, minimizing the possibility that other metabolites are involved, such as ubiquinone, for which kaempferol serves as a precursor (Berger et al, 2022), or terpenes and glucosinolates, the products of pathways that are interconnected with flavonoid metabolism (e.g., Sugimoto et al, 2021; Naik et al, 2023). It is worth noting that primary metabolites with strong antioxidant activity have also been implicated in control of ROS signaling into the clock (Kotchoni et al, 2009; Philippou et al, 2020; Román et al, 2021) but that little is yet known regarding the mechanism of action. Although the exact mechanism behind the modulation of circadian amplitude by flavonoids also remains unclear, our data point to the antioxidant role of dihydroxy B-ring flavonoids, potentially in chloroplasts, as the key to this relationship.

Signaling via $[Ca^{2+}]_{cyt}$ has also been implicated in the control of clock function in response to light and diverse stresses (Hotta et al, 2008; Martí Ruiz et al, 2018; Kidokoro et al, 2022). Although we did not observe changes in $[Ca^{2+}]_{cyt}$ in *tt4-2*, we did observe, in both *tt4-2* and *tt7-1*, a reduction in the $[Ca^{2+}]_{chl}$ spike that occurs at the light–dark transition (Figs 6B and S6A), a well-established phenomenon whose functional significance remains unknown (Martí Ruiz et al, 2018; Pivato et al, 2023; Kuang et al, 2024 *Preprint*). Interestingly, there is an emerging understanding of mechanisms connecting $Ca^{2+}$ and $H_2O_2$ signaling, not only in the cytoplasm (e.g., Bhattacharyya et al, 2025), but increasingly in chloroplasts. For example, elevated $[Ca^{2+}]_{chl}$ in response to treatment with oxidative stressors, including $H_2O_2$, has been reported in *Arabidopsis* cell cultures, in *Chlamydomonas*, and in the diatom, *Phaeodactylum tricornutum* (Sello et al, 2016; Pivato et al, 2023; Flori et al, 2024). This hints that the elevated $H_2O_2$ in flavonoid-deficient *Arabidopsis* seedlings could be the driver of altered $Ca^{2+}$ dynamics in the chloroplast. Flavonoids have been found at low levels within both the cytoplasm and chloroplasts (Agati et al, 2012; Winkel, 2019), raising the possibility that flavonoids could influence $[H_2O_2]_{chl}$ either directly or indirectly through targeted protein interactions, thereby altering $[Ca^{2+}]_{chl}$, which could then act through retrograde signaling to affect clock gene expression in the nucleus.

A key to understanding the mechanisms by which flavonoids influence the amplitude of *TOC1:LUC* cycling in vivo lies in

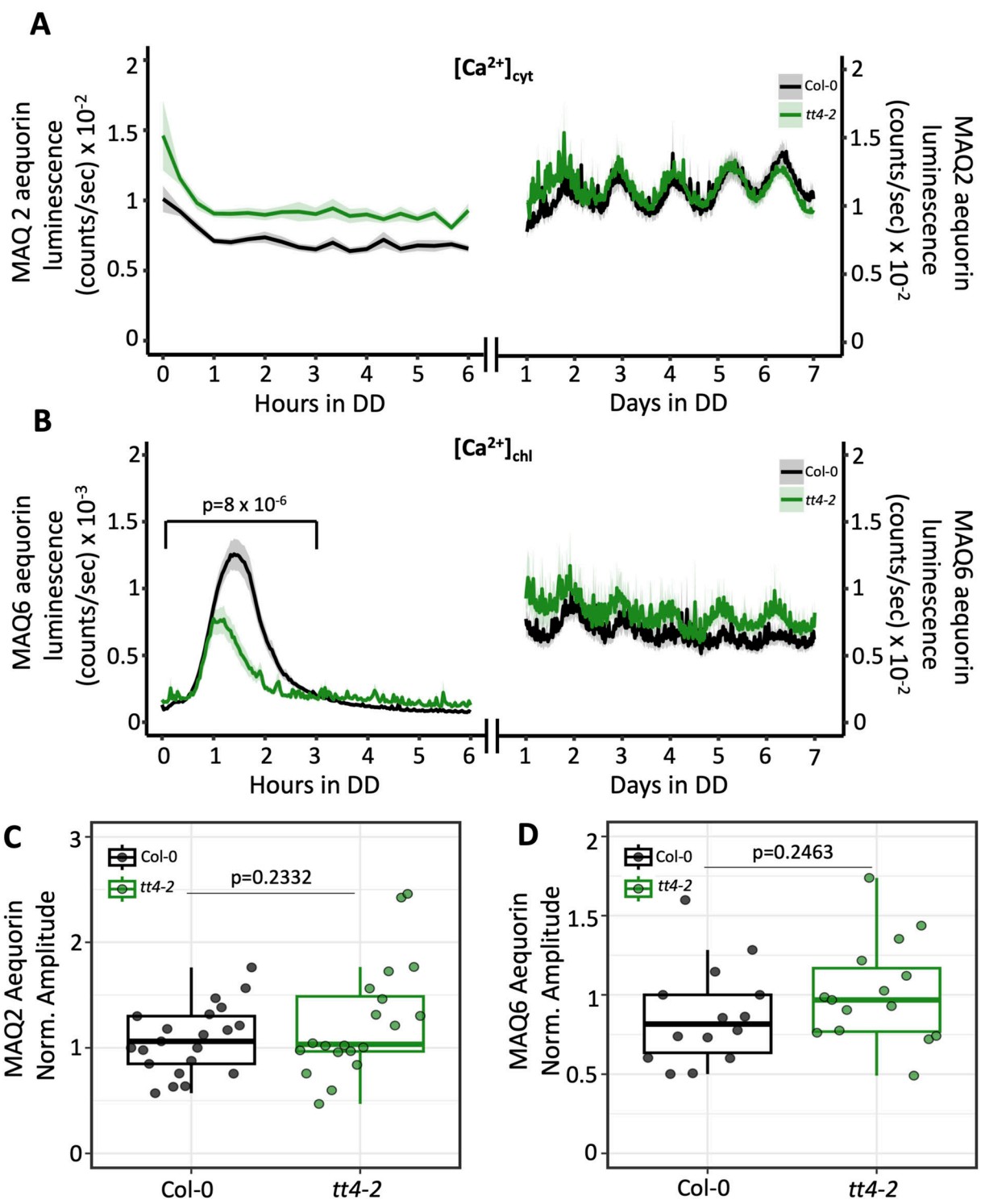

**Figure 6. Influence of flavonoid deficiency on Ca²⁺ levels in the chloroplast.**
**(A)** Bioluminescence output of cytosolic aequorin (MAQ2) in Col-0 (black) or *tt4-2* (green) (n = 20–21 biological replicates from two independent experiments). **(B)** Bioluminescence output of chloroplast-targeted (MAQ6) aequorin in Col-0 (black) or *tt4-2* (green) (n = 14–15 biological replicates from two independent experiments). *P*-value is calculated from two-way ANOVA comparing Col-0 and *tt4-2* between hours 0 and 3. **(C)** Amplitude of cytosolic aequorin rhythms in Col-0 or *tt4-2*, normalized to Col-0, which was set to 1. *P*-value is calculated from a two-tailed *t* test with unequal variance. **(D)** Amplitude of chloroplast-targeted aequorin rhythms in Col-0 or *tt4-2* normalized to Col-0, which was set to 1. *P*-value is calculated from a two-tailed *t* test with equal variance. In (C, D), boxplot midlines represent the median value (Q2); lower and upper lines represent 25th (Q1) and 75th percentiles (Q3), respectively. Whiskers represent the range of data within 1.5 interquartile range (Q3-Q1) from Q1 or Q3. Data in (A, B) represent the mean ± SEM.

identifying the specific flavonoid forms that are involved. Our data showed that most flavonoid glycosides exhibit little or no rhythmic accumulation in *Arabidopsis* seedlings, despite the strong circadian rhythmicity of flavonoid enzyme expression (Harmer et al, 2000; Nagel et al, 2015; Liebelt et al, 2019). This may be due to differences in flavonoid production across cell types or in subcellular localization that are not detected in whole seedling or leaf extracts, or reflect damping of the levels of individual forms because of turnover after reacting with ROS. Addressing this possibility will be another important goal of future efforts to characterize the functions of specific flavonoids in influencing circadian rhythmicity in plants.

# Materials and Methods

## Plant materials and growth conditions

*Arabidopsis* lines used in this study include the flavonoid mutants *tt4-11* (Salk_020583; Bowerman et al, 2012), a backcrossed version of *tt4-2* (Bennett et al, 2006), and *tt7-5* (Salk_053394; Bowerman et al, 2012) in the Col-0 ecotype, whereas *npr1-3 TOC1:LUC* (Cao et al, 1997; Zhou et al, 2016) and aequorin reporter lines MAQ2/MAQ6.8 (Knight et al, 1991; Lenzoni & Knight, 2018) were in the Col ecotype. The *tt7-1* mutant line was in the La-er ecotype (Koornneef, 1990). The *tt4-11* and *tt7-5* lines expressing *TOC1:LUC* were generated previously (Hildreth et al, 2022).

Seeds were surface-sterilized as described previously (Kubasek et al, 1992). For all experiments, seeds were sown on 0.8% agar containing 1x MS (Caisson Laboratories) with 2% sucrose, adjusted to pH 5.7 with 0.1 N KOH. Plates were sealed with Nesco film (Karlan Research Products Corp.) and stratified for 3–4 d at 4°C. Seedlings were then grown in 12-h light:12-h dark at 22°C under ~150 $\mu$E LED lights in an E-30B growth chamber (Percival). For experiments, plates within each desired genotype were selected at random for treatment groups. Experiments were not blinded.

For exogenous flavonoid and other chemical treatments, seeds were sown on agar plates containing 0.1% DMSO (control) or varying concentrations of naringenin, kaempferol, cyanidin (all from Sigma-Aldrich), quercetin (MP Biomedicals), or NPA (Neta Scientific) in 0.1% DMSO. KI treatment was performed by sowing seeds on agar plates containing water (control) or 100 $\mu$M KI (Sigma-Aldrich). For DPI treatment, seedlings were treated topically with 100 $\mu$l of 0, 1, 10, or 30 $\mu$M DPI (Sigma-Aldrich) in 0.1% DMSO at dusk on day 6 just before initiating luminescence recording.

## Luciferase reporter assay

Ten *TOC1:LUC* seeds were sown in a cluster onto 35-mm agar plates and grown in LD for 6 d as described above. For naringenin, DPI, and NPA treatment experiments, seeds were sown in clusters of 15. At dusk on day 5, seedlings were treated with 1 mM D-luciferin (potassium salt; GoldBio). Luminescence was recorded for 7 d starting with dusk on day 6 in constant darkness using a LumiCycle 32 luminometer (Actimetrics). Raw data from days 1 to

7 were baseline-detrended and processed using Fast Fourier Transform Non-Linear Least Squares (FFT NLLS) analysis in Bio-Dare2 (Zielinski et al, 2014) to determine amplitude and period. Rhythms that could not fit into a period between 18 and 30 h were excluded from amplitude or period quantifications.

## Intracellular $Ca^{2+}$ measurements

*tt4-2*, *tt7-1*, Col-0, and La-er were crossed with MAQ2 or MAQ6.8 aequorin reporter lines, visually screened for the flavonoid-deficient phenotypes at the F2 stage, and screened for the presence of the aequorin reporter at the F3 stage via polymerase chain reaction (PCR) (primers used for PCR are listed in Table S1) and luminescence measurements. F3 and F4 seeds homozygous for both the flavonoid mutation and the aequorin reporter gene were used for subsequent experiments. *tt4-11* and *tt7-5* mutant lines were not used to avoid T-DNA–mediated transcriptional silencing of the 35S promoter–driven aequorin transgene. $[Ca^{2+}]_{cyt}$ and $[Ca^{2+}]_{chl}$ rhythms were measured using seedlings expressing cytosolic (MAQ2) or chloroplast-targeted (MAQ6) aequorin reporter genes, respectively. For this, 20 seeds were sown onto 35-mm agar plates and grown as described above. At dusk on day 5, seedlings were treated with 10 $\mu$M coelenterazine (Invitrogen). Luminescence recordings began at dusk on day 6 for 7 d. We quantified amplitude and period of the rhythms as described above. Because of high noise in the $[Ca^{2+}]_{chl}$ rhythms, the rhythms were first smoothed in LumiCycle Analysis software (Actimetrics) using smooth 30 before amplitude quantification.

Estimation of the total aequorin present in transgenic seedlings was performed using the well-established method of discharging all reconstituted aequorin by applying an excess of $CaCl_2$ (Knight et al, 1996; Knight & Knight, 2000). Briefly, 6-d-old seedlings expressing the aequorin reporter were grown and treated with coelenterazine overnight as described above. Individual seedlings were placed into 1.5-ml tubes and then treated with 100 $\mu$l 1 M $CaCl_2$ and 10% ethanol to discharge all reconstituted aequorin. Luminescence was immediately measured for 10 sec in a 20/20n luminometer (Turner Biosystems).

## Hydrogen peroxide assays

For $H_2O_2$ measurements, 25–30 seeds were sown onto 35-mm agar plates and seedlings were grown in LD for 6 d as described above. Plates were kept under constant light or wrapped with aluminum foil for constant darkness starting on day 6 for 24 h before harvesting whole seedlings every 4 h for 24 h. For light/dark experiments, seedlings were harvested every 4 h for 24 h starting on day 7. For flavonoid/chemical treatments, seedlings were transferred to constant darkness on day 6 and collected after 24 h before harvesting seedlings every 4 h for 24 h. Collections in darkness were done under green LED light (~20 $\mu$E). Seedlings were weighed and snap-frozen in 1.5-ml microfuge tubes containing stainless steel beads (2.3 mm diameter; Small Parts) and kept frozen at –80°C. Frozen samples were ground to a powder using a TissueLyser II (QIAGEN) for three 10-sec pulses, submerging tubes in liquid nitrogen between pulses to prevent thawing. $H_2O_2$ levels were measured using the Amplex Red assay (Invitrogen) as

described elsewhere (Le et al, 2015; Chakraborty et al, 2016) with minor adjustments. Briefly, for each sample 200 $\mu$l of 50 mM sodium phosphate, pH 7.4, buffer was added per 30 mg of tissue. Samples were vortexed and then rocked at 4°C for 15 min before centrifugation at 15,871$g$, 4°C, for 5 min. The supernatant was moved to fresh 1.5-ml microfuge tubes and centrifuged again to remove excess debris. The resulting supernatant was diluted 1/5 with 50 mM sodium phosphate, pH 7.4, buffer, and 25 $\mu$l of diluted sample was added to 25 $\mu$l of Amplex Red working solution in a half-area black 96-well plate (Corning). The reaction was incubated at RT for 30 min, and fluorescence (545-nm excitation/590-nm emission) was measured with a BioTek Cytation 5 plate reader (Agilent). $H_2O_2$ content in pmol per mg fresh weight was calculated from absorbance values using a standard curve as described in the assay protocol provided by the manufacturer.

### LC-MS/MS analyses

For measurement of flavonoid accumulation patterns over time, samples were generated by sowing 25 sterilized seeds on 60-mm plates containing 1X MS, 2% sucrose overlaid with 30 $\mu$m nylon mesh (ELKO Filtering Co.) to prevent interference from agar contamination (Hildreth et al, 2020). Plates were incubated at 22°C with a 12-h/12-h LD cycle for 6 d. On day 7, collections began at dawn (zeitgeber time 0, indicating the onset of light) and continued every 4 h for 24 h. Seedlings were collected in 1.5-ml microfuge tubes containing two stainless steel beads (2.3 mm diameter; Small Parts), weighed, snap-frozen in liquid nitrogen, and stored at –80°C. Frozen samples were ground to a powder using a TissueLyser II as described above. The powdered samples were extracted in 200 $\mu$l methanol with 0.1% formic acid, vortexed, and sonicated for 5 min. The samples were then centrifuged at 13,000 rpm, 4°C, for 10 min, and 180 $\mu$l of the supernatant was transferred to a fresh 1.5-ml tube. The pelleted material was then re-extracted with 200 $\mu$l methanol, and the above process was repeated a second time. After pooling both 180 $\mu$l aliquots, the samples were dried and stored at –80°C.

Analyses were performed on a Shimadzu LC-MS 9030 QToF mass spectrometer interfaced with a LC-40B X3 UPLC, a SIL-40C X3 autosampler (10°C), and a CTO-40C column oven (40°C). A BEH C18 column (2.1 × 50 mm, 1.7-$\mu$m particle size; Waters) was used for chromatographic separation with solvent A (0.1% formic acid in water) and solvent B (0.1% formic acid in MeOH) at a flow rate of 0.4 ml min$^{-1}$. Solvent conditions began at 2% B and held for 1 min, then a linear gradient to 30% B at 5 min, and finally to 98% B at 10 min, which was held for 3 min. The gradient returned to starting conditions with a 0.5-min gradient to 2% B, followed by a 2.5-min hold. Sample injection volumes were 3 $\mu$l. Data were collected in both positive and negative modes in separate injections with MS scanning only. A master mix of all samples was prepared and analyzed by data-dependent acquisition (DDA) and/or MS/MS. Data processing was performed using MetaboAnalyst to obtain peak areas for metabolomic features (Pang et al, 2022). The resulting fragmentation patterns were matched to flavonoid glycosides in ReSpect and MassBank databases using MS-DIAL software (Horai et al, 2010; Sawada et al, 2012; Tsugawa et al, 2015). These compounds were confirmed as flavonoids by the presence

of an aglycone peak in the MS/MS spectra corresponding to the mass of kaempferol, quercetin, or isorhamnetin, whereas hexose and deoxyhexose glycosides were assigned based on neutral losses of 162 and 146, respectively (Supplemental Data 2).

For exogenous flavonoid treatments, seedlings were grown as described above on agar containing 0.1% DMSO as a control or 100 $\mu$M naringenin, kaempferol, cyanidin (all from Sigma-Aldrich), or quercetin (MP Biomedicals) in 0.1% DMSO. Seedlings were collected on day 7 at zeitgeber time 4. Metabolites were subsequently extracted and analyzed as described above.

### Statistical analyses

Statistical tests requiring multiple comparisons were performed using one-way ANOVA with Tukey's Honestly Significant Difference (HSD) post hoc test using the agricolae package in R with a significant cutoff of $P < 0.05$. Groups found to statistically differ are assigned different letters in order of highest mean, where a > b > c. Groups assigned ab are not statistically different from groups assigned a or b, groups assigned bc are not statistically different from groups assigned b or c, and groups assigned abc are not statistically different from groups assigned a, b, or c. Specific $P$-values from Tukey's post hoc tests are listed in Supplemental Data 1. Linear regression analysis was performed using the lm function in R. Correlation coefficient of the regression was reported as the adjusted $R^2$. To quantify rhythmicity of metabolites, LC-MS values were normalized to those at zeitgeber time 0, and rhythmicity parameters were quantified using MetaCycle with LS, ARS, and JTK algorithms (Wu et al, 2016). Rhythmic metabolites were defined as meta2d_$P < 0.05$.

## Data Availability

Raw LC-MS values and flavonoid MS/MS fragmentation patterns from the flavonoid circadian time course are listed in Supplemental Data 2. The LC-MS files for the exogenous flavonoid feeding experiments are available upon request.

## Supplementary Information

## Acknowledgements

This work was supported by grants from the National Science Foundation (grant IOB-0820674 to BSJ Winkel and GRFP Award DGE-2235205 to ES Littleton) and a Lay Nam Chang Dean's Discovery Grant from the College of Science at Virginia Tech (to BSJ Winkel and S Kojima). Seeds for the *tt4-2* backcrossed line, *TOC1:LUC*, *npr1-3 TOC1:LUC*, and pMAQ2/pMAQ6.8 were generous gifts from Gloria Muday (Wake Forest University), Robert McClung (Dartmouth College), Xinnian Dong (Duke University), and Marc Knight (Durham University), respectively. LC-MS experiments were performed at the Virginia Tech Mass Spectrometry Incubator (VT-MSI). The authors thank

Sam Chisholm for contributing to the initial analysis of rhythmicity in the flavonoid mass spectrometry dataset.

## Author Contributions

ES Littleton: conceptualization, data curation, formal analysis, validation, investigation, visualization, methodology, project administration, and writing—original draft, review, and editing.
SB Hildreth: data curation, formal analysis, validation, investigation, visualization, methodology, and writing—review and editing.
S Kojima: resources, formal analysis, supervision, funding acquisition, validation, visualization, methodology, and writing—review and editing.
BSJ Winkel: conceptualization, resources, data curation, formal analysis, supervision, funding acquisition, validation, visualization, methodology, project administration, and writing—original draft, review, and editing.

## Conflict of Interest Statement

The authors declare that they have no conflict of interest.

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
