## [Reviewer comments · Life Science Alliance]

Antioxidant properties of dihydroxy B-ring flavonoids modulate circadian amplitude in Arabidopsis

Evan Littleton, Sherry Hildreth, Shihoko Kojima, and Brenda Winkel

DOI: <https://doi.org/10.26508/lsa.202503328>

Corresponding author(s): Brenda Winkel, Virginia Tech

Review Timeline:

Submission Date:	2025-03-27
Editorial Decision:	2025-06-06
Revision Received:	2025-08-27
Editorial Decision:	2025-09-09
Revision Received:	2025-09-12
Accepted:	2025-09-15

Scientific Editor: Tim Fessenden

Transaction Report:

June 6, 2025

Re: Life Science Alliance manuscript #LSA-2025-03328-T

Brenda S.J. Winkel
Virginia Tech
Biological Sciences
Fralin Life Sciences Institute
1015 Life Science Circle (MC0477)
Blacksburg, Virginia 24061

Dear Dr. Winkel,

Thank you for submitting your manuscript entitled "Antioxidant properties of dihydroxy B-ring flavonoids modulate circadian amplitude in Arabidopsis" to Life Science Alliance. The manuscript was assessed by expert reviewers, whose comments are appended to this letter. We invite you to submit a revised manuscript addressing their main concerns as outlined below.

As you will see, reviewers overall appreciated the findings exploring the link between flavonoid metabolism and circadian signaling in Arabidopsis. Reviewer 1 made overall minor requests to refine the discussion, clarify methods, and improve figure clarity. Reviewer 2 similarly appreciated the main findings but expressed concerns over potential confounding effects of luciferase luminescence, and pointed to results on ROS signaling, the involvement of auxin, and calcium signaling that merited improved validation or altered discussion. We concur with this reviewer that confirming TOC expression changes without luciferase is essential to verify the main conclusions of this work, including observations on ROS levels. We also concur that luminescence results in Fig 6F must be calibrated or aequorin levels reported for these strains.

While you are revising your manuscript, please also attend to the below editorial points to help expedite the publication of your manuscript. Please direct any editorial questions to the journal office. The typical timeframe for revisions is three months. Please note that papers are generally considered through only one revision cycle, so strong support from the referees on the revised version is needed for acceptance.

When submitting the revision, please include a letter addressing the reviewers' comments point by point. We hope that the comments below will prove constructive as your work progresses.

Thank you for this interesting contribution to Life Science Alliance. We are looking forward to receiving your revised manuscript.

Sincerely,

-- Summary blurb (enter in submission system): A short text summarizing in a single sentence the study (max. 200 characters including spaces). This text is used in conjunction with the titles of papers, hence should be informative and complementary to the title and running title. It should describe the context and significance of the findings for a general readership; it should be

written in the present tense and refer to the work in the third person. Author names should not be mentioned.

B. MANUSCRIPT ORGANIZATION AND FORMATTING:

Reviewer #1 (Comments to the Authors (Required)):

The article entitled „Antioxidant properties of dihydroxy B-ring flavonoids modulate circadian amplitude in Arabidopsis" by Evan S. Littleton, Sherry B. Hildreth, Shihoko Kojima, Brenda S.J. Winkel addresses influence that secondary metabolites from flavonoid group play in circadian rhythmicity of plant metabolism.

This article is a direct continuation of findings from the same group published recently in the Plant Journal. In the previous work the authors found by transcriptomic methods that many genes in Arabidopsis that show rhythmic diurnal expression are affected by mutations in the flavonoid pathway gene - the chalcone synthase. T-DNA mutants in chalcone synthase gene have lower amplitude of diurnal expression of the major circadian regulator TOC and loose photosynthetic periodicity. In the current study the authors continue on elucidation of the mechanism behind that observation.

In the first experiment authors studied the effect on different flavonoids on the circadian parameters of TOC expression by using a previously published reporter line in which TOC promoter drives the luciferase expression. They showed that the high concentrations of naringenin, one of the initial products of the flavonoid pathway is decreasing the amplitude of TOC expression in WT. This situation mimics biochemically the accumulation of naringenin which should be the biochemical effect of the tt7 mutation. Surprisingly, in the previous report in the tt7 mutant the TOC amplitude of expression was increasing both after transfer from 12h darkness/12h light cycle or LL conditions to darkness (Plant Journal publication, fig. 4), also current report Fig. 1C. I would like the authors to explain this discrepancy.

I also have to ask for introducing the scheme for chemical reactions with names of enzymes, compounds and mutants marked on the scheme- just like the fig. 1A in the previous report as Fig.1. It will make the understanding of the experiments much easier. It is not stated what are the cellular concentrations of the compounds used to feed the plants- are the experimentally used conditions relevant to the cytoplasmic concentrations? - please introduce the data in the text.

Next the authors aimed at showing if the flavonoids show diurnal accumulation in plants by LC-MS. In fact only two did. The situation is clearly described in the text, but the Fig.2 is slightly misleading- maybe the two compounds that indeed have the p values statistically significant should be more distinguished on the panel? The authors gave the full initial data from MS experiments in the supplement, but did not showed the spectra or patterns of fragmentation for the studied compounds- where they easy to identify- they have very similar structures and MW, please comment in the text.

Later the authors study the effect of flavonoids on accumulation of hydrogen peroxide in the cells. On Fig.4 it is not clear to me, to what do the p values correspond- is the amount of hydrogen peroxide in Col compared to the mutants at one time point- which? Or is the periodicity compared? Or amplitude? It is not clearly stated. The data shown for peroxide accumulation are nicely consistent with the data on TOC:LUC activity shown on Fig.1 for all experimental variants. Also the experiments for chemical manipulation of ROS levels in flavonoid-deficient plants shown on Fig.5 are convincing.

Unfortunately the further explanation of the mechanism of actions of flavonoids is not that clear. The cytosolic calcium ions level is not changed (Fig. 6A,B), also the changes of intrachloroplast free calcium are low (Fig. 6 C,D,E), interestingly the calcium spike after the onset of darkness is reduced in the tt4 mutant, but this observation is not widely discussed. The involvement of auxin signalling or NPR receptor was excluded.

Summarizing the current report shows some new and interesting data on the involvement of flavonoids in the regulation of TOC expression. The data on TOC expression in response to flavonoid application, the rhythmicity of some flavonoid accumulation and hydrogen peroxide accumulation are consistent. The discrepancy between the chemical and mutational approach must be cleared and more widely approached in the discussion. Unfortunately the further mechanism of action on flavonoids on circadian rhythm is still to be discovered.

The article is very nicely written and illustrated in a consistent way. The statistical analysis was performed whenever possible. The one thing that is missing for me, that would make the text much easier to follow is the reaction scheme with all the enzymes, compounds and mutant names labeled.

Reviewer #2 (Comments to the Authors (Required)):

General points:

This builds on the interesting link between flavonoid biosynthesis/levels and the circadian clock in plants. The work focuses primarily on a mutation in the chalcone synthase gene, which results in reduced flavonoid content in plants.

- The strengths of the work lie in rigorous experimentation and data cataloguing, and pursuing a very interesting hypothesis;
- The limitations lie in the use of a single reporter gene as an output and the significant descriptive data and speculation. A less major limitation is that while amplitude is the focus, the effects on circadian period are only briefly addressed.

Points for authors to consider:

1. This study measures the expression of TOC1 using a promoter fusion to firefly luciferase and measures luminescent activity, an established method. However, as this is a proxy for measuring the expression of this gene, other factors, such as luciferase activity, can have a non-specific effect on the output. For the main conclusion that flavonoid levels truly regulate the amplitude of TOC1 expression through alterations in reactive oxygen species levels to be valid, it is important that the native gene transcript is also measured. Direct qRT-PCR of native TOC1 mRNA would reinforce the luciferase findings, particularly under flavonoid or ROS-altered conditions. Alternatively, it would be beneficial to conduct appropriate controls to measure the impact of altered levels of reactive oxygen species, flavonoids, etc. on luciferase activity itself, for example by examining the effects on a modestly expressed constitutive promoter fused to firefly luciferase. Alternatively one could test clock amplitude changes via additional clock gene reporters, such as CCA1::LUC or LHY::LUC to ensure the phenomenon is TOC1-specific.
2. Figure 1 shows a clear effect on the amplitude of luciferase activity circadian oscillations upon flavonoid treatment, such as naringenin, but there is also a general increase in total mass luciferase activity under these conditions. The authors do not discuss this, but could the increase in amplitude simply be due to an increase in signal-to-noise ratio, i.e. could flavonoids potentiate the output of the luminescent signal of firefly luciferase, making everything brighter and increasing the signal-to-noise ratio, thereby measuring higher amplitude oscillations for technical rather than biological reasons? In this regard, the controls I suggested above would be very important.
3. Figure 2 shows a very careful, detailed analysis of the levels of different flavonoids at different circadian times. This is a very interesting piece of information and strongly supports the idea of the role of flavonoids in circadian biology.
4. The data presented in Figure 3 and the supplementary data are based on using NPA to inhibit auxin transport. The rationale is that, since it is known that flavonoids inhibit auxin transport *in vivo*, the mutant used by the authors should show increased auxin activity/transport, which could explain the changes in apparent TOC1 expression observed. The authors are using NPA in an attempt to reverse the potential release of auxin transport due to the reduced levels of flavonoids in the mutant. However, the conclusion that inhibiting auxin transport in the mutant background has no effect is consistent with the idea that the effects on TOC1 expression are not due to auxin. Nevertheless, auxin transport is a complex phenomenon involving many different transport proteins, so it could be that NPA does not faithfully mimic this specific change that might occur in response to flavonoid changes. Nevertheless, more muted conclusions could be drawn. It would be important to test the effects of adding auxins themselves e.g. IAA to wild type to complement this work with inhibitors. In any case, I believe the authors need to provide further answers to address the discussion regarding auxin transport and the conclusions drawn from this study.
5. The data presented in Figure 4 show some very thorough work investigating the levels of reactive oxygen species and their correlation with TOC1::LUC expression. The correlation is clear. However, one is once again left wondering whether this correlation is due to luciferase activity itself as opposed to the transcription of the TOC1 promoter specifically. A control to measure native TOC1 transcript would be useful.
6. The data in Figure 5, which uses the respiratory burst oxidase homologue inhibitor DPI, seems to indicate a relationship between DPI-dependent reactive oxygen species and the level of TOC1::LUC expression. However, it is far from clear that flavonoids would stimulate the production of reactive oxygen species through concerted activation of respiratory burst oxidases; rather, they operate by behaving as quenching buffers for reactive oxygen species produced through various sources.
7. Lenzoni and Knight (2018) are quoted as describing chloroplast calcium elevations in response to a light-to-dark transition, but in fact they are describing responses to high temperatures.

8. The authors postulate retrograde signalling between the nucleus and the chloroplast, but there is no foundation for this given the location of flavonoids and no clear indication of a signal that communicates between these two organelles. In my opinion, this discussion could do with reframing or limiting this speculation to a more cautious statement.

9. The data in Figure 6 show that there is little difference in circadian calcium oscillations between wild-type and mutant plants, in both the cytosol and the chloroplast. It is unclear why these experiments were performed in complete darkness after entrainment, as this is not in line with previous studies and makes comparison difficult. Cytosolic calcium oscillations appear robust in constant darkness, which contrasts with previous findings by others. Indeed, they do not dampen at all. In contrast to previous studies, where circadian oscillations in the chloroplast are NOT observed, the authors observe them and they appear to mirror the cytosolic oscillations and have the same timing. Could the chloroplast lines be ineffectively targeted, with some aequorin contamination in the cytosol?

10. Figure 6F shows that less calcium-dependent luminescence is produced in response to the light-to-dark transition in mutant lines expressing aequorin in the chloroplast. The authors conclude that this indicates a reduced calcium response; however, this data is not calibrated, representing raw luminescence, and may simply be due to the mutant line having lower aequorin expression. The data must be calibrated, or at least an estimate of aequorin expression must be made and compared between the wild type and the mutant, before this conclusion can be drawn. As this is a vital conclusion, this control is essential.

Overall: this study is a valuable contribution to our understanding of the relationship between flavonoid biochemistry and circadian regulation in plants. However, the mechanistic conclusions would be greatly strengthened by a more rigorous assessment of native gene expression and luciferase-independent controls. Addressing concerns (even through more cautious writing) about ROS specificity, auxin independence and the validity of calcium signalling would also improve this manuscript.

RESPONSE TO REVIEWERS

Please note that all changes to the manuscript described below refer to page numbers in the accompanying revised file, where these modifications/additions are highlighted in blue text.

Reviewer #1 (Comments to the Authors (Required)):

The article entitled „Antioxidant properties of dihydroxy B-ring flavonoids modulate circadian amplitude in Arabidopsis" by Evan S. Littleton, Sherry B. Hildreth, Shihoko Kojima, Brenda S.J. Winkel addresses influence that secondary metabolites from flavonoid group play in circadian rhythmicity of plant metabolism.

This article is a direct continuation of findings from the same group published recently in the Plant Journal. In the previous work the authors found by transcriptomic methods that many genes in Arabidopsis that show rhythmic diurnal expression are affected by mutations in the flavonoid pathway gene - the chalcone synthase. T-DNA mutants in chalcone synthase gene have lower amplitude of diurnal expression of the major circadian regulator TOC and loose photosynthetic periodicity. In the current study the authors continue on elucidation of the mechanism behind that observation.

In the first experiment authors studied the effect on different flavonoids on the circadian parameters of TOC expression by using a previously published reporter line in which TOC promoter drives the luciferase expression. They showed that the high concentrations of naringenin, one of the initial products of the flavonoid pathway is decreasing the amplitude of TOC expression in WT. This situation mimics biochemically the accumulation of naringenin which should be the biochemical effect of the tt7 mutation. Surprisingly, in the previous report in the tt7 mutant the TOC amplitude of expression was increasing both after transfer from 12h darkness/12h light cycle or LL conditions to darkness (Plant Journal publication, fig. 4), also current report Fig. 1C. I would like the authors to explain this discrepancy.

We believe our results are not contradictory, because as can be seen in Figure EV1, the biochemical effect of the *tt7* mutation is NOT to accumulate naringenin, but rather to shunt flux into the monohydroxy branch pathway, leading to accumulation of kaempferol. A lack of naringenin accumulation in *tt7* has been confirmed by other groups (see figure 4 in Gayomba & Muday, Development 2020). Thus there is no discrepancy between the biochemical and genetic results: we show that naringenin treatment lowers *TOC1:LUC* amplitude in *tt4* (Fig. 1A, B) by rescuing dihydroxy B-ring flavonoid production (Fig. EV1), but not in *tt7* due to the absence of F3'H activity. These data therefore demonstrate that dihydroxy B-ring flavonoids are responsible for elevated *TOC1:LUC* amplitude.

I also have to ask for introducing the scheme for chemical reactions with names of enzymes, compounds and mutants marked on the scheme- just like the fig. 1A in the previous report as Fig.1. It will make the understanding of the experiments much easier.

A schematic of the flavonoid pathway is presented as Figure EV1, but can easily be incorporated into the main text as Figure 1, should the editor recommend we do so.

It is not stated what are the cellular concentrations of the compounds used to feed the plants- are the experimentally used conditions relevant to the cytoplasmic concentrations? – please introduce the data in the text.

We appreciate the reviewer's concern. The concentrations we used in the current study are based on standard methods established and reported by other groups and ours over many years, and several of these are cited in the manuscript (e.g., Buer and Muday, Plant Cell 2004). The exact concentration of flavonoids *in vivo* is difficult to quantify as the accumulation of flavonoids varies greatly across developmental stages, tissues, and organelles. However, to address the reviewer's concern we used LC-MS/MS to measure relative levels of flavonoids in 7-day-old whole-seedling extracts in both wild-type and mutant lines grown on exogenous flavonoids under the same conditions used to examine effects on *TOC1:LUC* expression [new Figure EV4, as well as descriptions in the Results (new paragraph, bottom of page 3) and Methods sections (blue text, pp. 16 & 17)]. Our data show that exogenous supplementation of *tt4* with 100 μ M naringenin resulted accumulation of kaempferol, quercetin, and isorhamnetin glycosides. In *tt7*, growth on 100 μ M naringenin further increased the already-elevated levels of kaempferol glycosides, but did not rescue the production of quercetin or isorhamnetin glycosides, as expected, due to absence of the F3'H activity needed to produce these compounds (Figure EV1). Kaempferol, quercetin, and cyanidin treatment led to only slight increases in these compounds in any of the lines, including wild type, likely due to their known limited transport to above-ground tissues (Buer *et al.*, Plant Physiol. 2007). For simplicity, chromatograms were shown to visualize relative differences between genotypes and treatments; the differences were not quantified as the large variation in flavonoid glycoside levels across samples exceeded the dynamic range of the mass spectrometer. Together, these findings confirm that the long-standing method of treatment of *Arabidopsis* seedlings with exogenous flavonoids at 100 μ M concentrations indeed results in flavonoid accumulation at physiologically-relevant levels, comparable to those present in untreated counterparts.

Next the authors aimed at showing if the flavonoids show diurnal accumulation in plants by LC-MS. In fact only two did. The situation is clearly described in the text, but the Fig.2 is slightly misleading- maybe the two compounds that indeed have the p values statistically significant should be more distinguished on the panel? The authors gave the full initial data from MS experiments in the supplement, but did not showed the spectra or patterns of fragmentation for the studied compounds- where they easy to identify- they have very similar structures and MW, please comment in the text.

We agree that the flavonoids with statistically-significant rhythmic accumulation could be better differentiated in Figure 2. We have now highlighted those significant p-values in red to indicate their significance. Regarding the identification of flavonoids from our LC-MS/MS data, we included the MS/MS fragmentation pattern of the identified flavonoids in the

supplemental data sheet containing the LC-MS data, therefore we do not feel it is necessary to show images of the MS/MS spectra. However, to improve the clarity of the data analysis we have included additional information in the LC-MS/MS section of the methods discussing how flavonoids were identified. We also further clarified the method of data normalization in the legend for Figure 2.

Later the authors study the effect of flavonoids on accumulation of hydrogen peroxide in the cells. On Fig.4 it is not clear to me, to what do the p values correspond- is the amount of hydrogen peroxide in Col compared to the mutants at one time point- which? Or is the periodicity compared? Or amplitude? It is not clearly stated. The data shown for peroxide accumulation are nicely consistent with the data on TOC:LUC activity shown on Fig.1 for all experimental variants. Also the experiments for chemical manipulation of ROS levels in flavonoid-deficient plants shown on Fig.5 are convincing.

We have added information in the legend for Figure 4 to better describe the statistical tests performed.

Unfortunately the further explanation of the mechanism of actions of flavonoids is not that clear. The cytosolic calcium ions level is not changed (Fig. 6A,B), also the changes of intrachloroplast free calcium are low (Fig. 6 C,D,E), interestingly the calcium spike after the onset of darkness is reduced in the tt4 mutant, but this observation is not widely discussed. The involvement of auxin signalling or NPR receptor was excluded.

Together these experiments point to the novel possibility that chloroplasts may be a site of action for flavonoids, but we are being deliberately cautious in our interpretations of the data at this stage. We have now added new information on the effects of the *tt7-1* mutation on intrachloroplast calcium levels as further evidence for this possibility (Figure EV6; text highlighted in blue, bottom of page 11).

Summarizing the current report shows some new and interesting data on the involvement of flavonoids in the regulation of TOC expression. The data on TOC expression in response to flavonoid application, the rhythmicity of some flavonoid accumulation and hydrogen peroxide accumulation are consistent. The discrepancy between the chemical and mutational approach must be cleared and more widely approached in the discussion. Unfortunately the further mechanism of action on flavonoids on circadian rhythm is still to be discovered.

As explained above, there is no discrepancy between the chemical and mutational approach. The different effects of *tt4* and *tt7* on flavonoid metabolism (Fig. EV1) are clearly defined, both in the text (first paragraph of the Results, p. 2) and in Fig. EV1. It is this difference that allows us to pinpoint dihydroxy B-ring flavonoids as the key effectors of *TOC1* promoter activity and the effects on Ca²⁺ homeostasis (Results: final sentence of first paragraph on page 2, first and second paragraphs on p. 3, second and third paragraphs on p. 7, third and fourth paragraphs on page 11). We agree that the mechanism of action is not fully clear at this point, as emphasized in our original submission of the manuscript (see

especially the Introduction (final sentence, p. 2), and the Discussion (end of second paragraph, p. 13).

The article is very nicely written and illustrated in a consistent way. The statistical analysis was performed whenever possible. The one thing that is missing for me, that would make the text much easier to follow is the reaction scheme with all the enzymes, compounds and mutant names labeled.

We appreciate the reviewer's helpful comments and their careful reading of this manuscript, also in light of our previous report on this subject. Please see the response above regarding the flavonoid pathway schematic.

Reviewer #2 (Comments to the Authors (Required)):

General points:

This builds on the interesting link between flavonoid biosynthesis/levels and the circadian clock in plants. The work focuses primarily on a mutation in the chalcone synthase gene, which results in reduced flavonoid content in plants.

- The strengths of the work lie in rigorous experimentation and data cataloguing, and pursuing a very interesting hypothesis;*
- The limitations lie in the use of a single reporter gene as an output and the significant descriptive data and speculation. A less major limitation is that while amplitude is the focus, the effects on circadian period are only briefly addressed.*

Points for authors to consider:

1. This study measures the expression of TOC1 using a promoter fusion to firefly luciferase and measures luminescent activity, an established method. However, as this is a proxy for measuring the expression of this gene, other factors, such as luciferase activity, can have a non-specific effect on the output. For the main conclusion that flavonoid levels truly regulate the amplitude of TOC1 expression through alterations in reactive oxygen species levels to be valid, it is important that the native gene transcript is also measured. Direct qRT-PCR of native TOC1 mRNA would reinforce the luciferase findings, particularly under flavonoid or ROS-altered conditions. Alternatively, it would be beneficial to conduct appropriate controls to measure the impact of altered levels of reactive oxygen species, flavonoids, etc. on luciferase activity itself, for example by examining the effects on a modestly expressed constitutive promoter fused to firefly luciferase. Alternatively one could test clock amplitude changes via additional clock gene reporters, such as CCA1::LUC or LHY::LUC to ensure the phenomenon is TOC1-specific.

This is an important point. In our previous manuscript (Hildreth et al., 2022) we confirmed that there is a direct relationship between alterations in *TOC1* transcript levels (determined by qRT-PCR) in flavonoid mutant lines relative to wild type, with *TOC1:LUC* amplitude as assessed by luminescence measurements (Figs. 3B and 4). This is stated in the Introduction

(at the start of the final paragraph, p. 2). In the same earlier manuscript, we showed that the amplitude of another luciferase reporter, *CCA1:LUC*, is unchanged in the *tt4* and *tt7* mutant lines (Fig. 4); a statement to this effect has now been added to the Results (first paragraph, blue text, p. 2). These prior findings show that the responses of the *TOC1:LUC* reporter described in this manuscript are not merely the result of altered luciferase activity due to the lack of endogenous flavonoids in the mutant lines.

We will note that promoter:luciferase fusions, including *TOC1:LUC* and *CCA1:LUC*, have been used extensively as reporters of plant clock activity under different ROS conditions, including situations where chemical treatments or mutant backgrounds altered ROS levels and had no effect on luminescent output (e.g. Lai et al., PNAS 2012; Philippou et al. Front. Physiol. 2020; Phan et al. FEBS Lett. 2022; Paeng et al., Molecular Plant, 2025). Although we have not tested the effects of ROS on luciferase activity ourselves, these data strongly support the conclusion that elevated ROS conditions alters the transcription of luciferase reporters in a promoter-specific manner.

2. Figure 1 shows a clear effect on the amplitude of luciferase activity circadian oscillations upon flavonoid treatment, such as naringenin, but there is also a general increase in total mass luciferase activity under these conditions. The authors do not discuss this, but could the increase in amplitude simply be due to an increase in signal-to-noise ratio, i.e. could flavonoids potentiate the output of the luminescent signal of firefly luciferase, making everything brighter and increasing the signal-to-noise ratio, thereby measuring higher amplitude oscillations for technical rather than biological reasons? In this regard, the controls I suggested above would be very important.

This is another important point, and we actually do have good evidence that flavonoids do not affect the luminescent signal directly. As now underscored in the first paragraph of the Results (p. 2), our earlier work showed that the amplitude of *CCA1:LUC* reporter activity was unchanged between WT and the flavonoid mutant lines (Hildreth et al., 2022). Thus the effect of flavonoids appears to be specific to *TOC1:LUC* promoter activity, and is not a general effect on the luciferase/luciferin signal.

3. Figure 2 shows a very careful, detailed analysis of the levels of different flavonoids at different circadian times. This is a very interesting piece of information and strongly supports the idea of the role of flavonoids in circadian biology.

We appreciate the reviewer's comment.

4. The data presented in Figure 3 and the supplementary data are based on using NPA to inhibit auxin transport. The rationale is that, since it is known that flavonoids inhibit auxin transport in vivo, the mutant used by the authors should show increased auxin activity/transport, which could explain the changes in apparent TOC1 expression observed. The authors are using NPA in an attempt to reverse the potential release of auxin transport due to the reduced levels of flavonoids in the mutant. However, the conclusion that inhibiting auxin transport in the mutant

background has no effect is consistent with the idea that the effects on TOC1 expression are not due to auxin. Nevertheless, auxin transport is a complex phenomenon involving many different transport proteins, so it could be that NPA does not faithfully mimic this specific change that might occur in response to flavonoid changes. Nevertheless, more muted conclusions could be drawn. It would be important to test the effects of adding auxins themselves e.g. IAA to wild type to complement this work with inhibitors. In any case, I believe the authors need to provide further answers to address the discussion regarding auxin transport and the conclusions drawn from this study.

We agree that auxin transport is indeed a very complex phenomenon and that we should be more cautious in drawing conclusions from the NPA experiments. We do already have a detailed explanation of the mechanism of flavonoid and NPA action on auxin transport. To temper the conclusions, we have added “appear to” in the subtitle for this section of the Results (p. 6) and have rewritten the final sentence of this section, including using the term “suggest” rather than “indicate,” to clarify the interpretation (end of first paragraph on p. 6). To this same end, we have also made a small modification to the Abstract (p. 1).

5. The data presented in Figure 4 show some very thorough work investigating the levels of reactive oxygen species and their correlation with TOC1::LUC expression. The correlation is clear. However, one is once again left wondering whether this correlation is due to luciferase activity itself as opposed to the transcription of the TOC1 promoter specifically. A control to measure native TOC1 transcript would be useful.

Please see our response to comment #1.

6. The data in Figure 5, which uses the respiratory burst oxidase homologue inhibitor DPI, seems to indicate a relationship between DPI-dependent reactive oxygen species and the level of TOC1::LUC expression. However, it is far from clear that flavonoids would stimulate the production of reactive oxygen species through concerted activation of respiratory burst oxidases; rather, they operate by behaving as quenching buffers for reactive oxygen species produced through various sources.

We would like to clarify that we did not use DPI to examine the mechanism of action of flavonoids, which we agree does not involve stimulating the production of ROS. Rather, we used DPI to reduce ROS production and determine whether this affected *TOC1::LUC* expression differentially in the presence (wild type) or absence (*tt4*) of flavonoids. In Fig. 5, we show that potassium iodide (KI), an H₂O₂ scavenger that may behave similarly to flavonoids in quenching reactive oxygen species (I⁻ reacts with H₂O₂ to generate I₂ and H₂O), also reduces the amplitude of *TOC1::LUC* in *tt4*. The use of these two chemicals indicates that the enhanced *TOC1::LUC* amplitude in *tt4* can be reduced by lowering ROS levels regardless of the mechanism (preventing ROS production with DPI or scavenging excess H₂O₂ with KI). We have clarified this by adding additional information regarding the mechanism of H₂O₂ scavenging by KI in this section of the Results (p. 9).

7. Lenzi and Knight (2018) are quoted as describing chloroplast calcium elevations in response to a light-to-dark transition, but in fact they are describing responses to high temperatures.

That reference is indeed incorrectly cited in this one location. This citation has now been deleted and replaced by two others, to Wood et al., Plant Physiol. 2001 and Johnson et al., Science 1995, in its place.

8. The authors postulate retrograde signalling between the nucleus and the chloroplast, but there is no foundation for this given the location of flavonoids and no clear indication of a signal that communicates between these two organelles. In my opinion, this discussion could do with reframing or limiting this speculation to a more cautious statement.

We agree that the involvement of flavonoids in this process is purely speculative. We have tempered the associated statement in the abstract; the other two statements, in the Results (top of p. 12) and Discussion (top of p.14), are already quite cautious. Therefore we do not feel the need to revise.

9. The data in Figure 6 show that there is little difference in circadian calcium oscillations between wild-type and mutant plants, in both the cytosol and the chloroplast. It is unclear why these experiments were performed in complete darkness after entrainment, as this is not in line with previous studies and makes comparison difficult. Cytosolic calcium oscillations appear robust in constant darkness, which contrasts with previous findings by others. Indeed, they do not dampen at all. In contrast to previous studies, where circadian oscillations in the chloroplast are NOT observed, the authors observe them and they appear to mirror the cytosolic oscillations and have the same timing. Could the chloroplast lines be ineffectively targeted, with some aequorin contamination in the cytosol?

The aequorin measurements were conducted in constant darkness to enable its luminescence detection using our luminometer. This experimental design also maintained the growth conditions identical to those used for the *TOC1:LUC* luminescence experiments. These same reporters have been used in several other studies to examine cytosolic ($[Ca^{2+}]_{cyt}$) and chloroplast ($[Ca^{2+}]_{chl}$) rhythms in various conditions and species (Johnson et al., Science 1995, Wood et al., Plant Physiol. 2001, Martí Ruiz et al., New Phytol. 2019). While it is true that $[Ca^{2+}]_{cyt}$ rhythms quickly dampen in constant darkness in the absence of sucrose, they can be sustained in the presence of sucrose (Fig. 8 in Martí Ruiz et al., New Phytol. 2019). Because of this, we included 2% sucrose in the medium in all our of experiments and detected robust $[Ca^{2+}]_{cyt}$ rhythms in constant darkness. Furthermore, $[Ca^{2+}]_{chl}$ rhythms in tobacco seedlings were previously shown to be arrhythmic in constant light but sustained in constant darkness (Fig. 4 in Johnson et al., Science 1995 and Fig. 1 in Wood et al., Plant Physiol. 2001), consistent with our observation of $[Ca^{2+}]_{chl}$ rhythms in constant darkness.

Regarding the similar timing between cytosolic and chloroplast calcium oscillations, this is also seen in the literature, such as Fig. 1 in Wood et al. (Plant Physiol. 2001), in which oscillations in cytosolic and chloroplast calcium rhythms were both rhythmic and with a similar phase (i.e. timing) in constant darkness, corroborating our findings as well.

As for the possibility of ineffective targeting of the chloroplast-targeted aequorin, 93.8% of the aequorin was localized to chloroplasts when the same reporter was used in tobacco protoplasts (Johnson et al., Science 1995), and we expect this to be similar in *Arabidopsis*. Furthermore, the large spike we detected in $[Ca^{2+}]_{chl}$ measurements after transfer to constant darkness (Fig. 6F) was not detected in our $[Ca^{2+}]_{cyt}$ measurements, consistent with previous reports (e.g., Fig. 4 in Johnson et al., Science 1995 and Fig. 1 in Wood et al., Plant Physiol 2001). This indicates that the majority of the signal when measuring $[Ca^{2+}]_{chl}$ is coming from chloroplast-targeted aequorin. To address this, we have revised Fig. 6A to display the lack of a spike in $[Ca^{2+}]_{cyt}$ after transfer to constant darkness. For clarification, we also combined panels C and F in the original Fig. 6, as both luminescence measurements were from the same experiment at different times in constant darkness (hours 0-6 or days 1-7).

10. Figure 6F shows that less calcium-dependent luminescence is produced in response to the light-to-dark transition in mutant lines expressing aequorin in the chloroplast. The authors conclude that this indicates a reduced calcium response; however, this data is not calibrated, representing raw luminescence, and may simply be due to the mutant line having lower aequorin expression. The data must be calibrated, or at least an estimate of aequorin expression must be made and compared between the wild type and the mutant, before this conclusion can be drawn. As this is a vital conclusion, this control is essential.

We agree that this is an important control. To address this, we estimated the total amount of reconstituted aequorin by measuring the luminescence of individual seedlings expressing aequorin after treatment with an excess (1M) of $CaCl_2$ in 10% ethanol, which discharges all reconstituted aequorin in the sample as previously reported (Knight et al. Plant Cell, 1996, Knight and Knight, J. Exp. Bot., 2000). Our results show that there is no statistical difference in luminescence between Col-0 and *tt4-2* seedlings after discharging all aequorin, indicating the difference in aequorin luminescence after the transition between WT and mutant is not due to intrinsic differences in total aequorin expression. We also show that the *tt7-1* mutant in the Landsberg erecta (La-er) ecotype has a decreased light-dark $[Ca^{2+}]_{chl}$ spike relative to La-er wildtype (Fig. EV6C), suggesting that the decreased $[Ca^{2+}]_{chl}$ response to light-dark transition is not genotype-specific, but rather caused by a loss of dihydroxy B-ring flavonoids. Similar to *tt4*, this decreased signal was also not due to lower aequorin expression, as discharging all chloroplast-targeted aequorin was slightly higher in *tt7-1* relative to La-er (Figure EV6B). We have included these new data in Figure EV6 and added related information to the Results (third paragraph, p. 11), Discussion (first and third paragraphs, p. 13) and Methods (p. 15).

Overall: this study is a valuable contribution to our understanding of the relationship between flavonoid biochemistry and circadian regulation in plants. However, the mechanistic conclusions would be greatly strengthened by a more rigorous assessment of native gene expression and luciferase-independent controls. Addressing concerns (even through more cautious writing) about ROS specificity, auxin independence and the validity of calcium signalling would also improve this manuscript.

We thank the reviewer's very helpful and thoughtful comments.

We believe we have addressed all of the reviewers' concerns and that the manuscript is now ready for publication.

September 9, 2025

RE: Life Science Alliance Manuscript #LSA-2025-03328-TR

Dr. Brenda S.J. Winkel
Virginia Tech
Biological Sciences
Fralin Life Sciences Institute
1015 Life Science Circle (MC0477)
Blacksburg, Virginia 24061

Dear Dr. Winkel,

Thank you for submitting your revised manuscript entitled "Antioxidant properties of dihydroxy B-ring flavonoids modulate circadian amplitude in Arabidopsis". As you will see, Reviewer 1 is now satisfied and recommends publication. We feel the revised manuscript suitably addressed the concerns of Reviewer 1, who was unfortunately unavailable to re-review. We would be happy to publish your paper in Life Science Alliance pending final revisions necessary to meet our formatting guidelines.

- Please upload your Table in editable .doc or Excel format.
- LSA allows supplementary figures, but not EV Figures; please update your callouts for the Supplementary Figures in the manuscript Fig EV1A = Fig S1A). The same applies to the tables as well.
- Please add ORCID ID for secondary corresponding author--they should have received instructions on how to do so.
- Please add the X and Bluesky handles of your host institute/organization, as well as your own and/or one of the authors, in our system.
- Please move your main, supplementary figure, and table legends to the main manuscript text after the references section.
- Please add an Author Contributions section to your main manuscript text.

LSA now encourages authors to provide a 30-60 second video where the study is briefly explained. We will use these videos on social media to promote the published paper and the presenting author (for examples, see <https://docs.google.com/document/d/1-UWCfbE4pGcDdcgzcmiuJl2XMBJnxKYeqRvLLrLSo8s/edit?usp=sharing>). Corresponding or first-authors are welcome to submit the video. Please submit only one video per manuscript. The video can be emailed to contact@life-science-alliance.org

A. FINAL FILES:

B. MANUSCRIPT ORGANIZATION AND FORMATTING:

Thank you for your attention to these final processing requirements. Please revise and format the manuscript and upload materials as soon as you are able.

Sincerely,

Reviewer #1 (Comments to the Authors (Required)):

Comments to authors:

Thank you for a new revised version of the manuscript entitled: „Antioxidant properties of dihydroxy B-ring flavonoids modulate circadian amplitude in Arabidopsis".

Concerning the answers to my questions:

Q1 The flow of the metabolites in the tt7 mutant was explained in detail.

Q2 The schematic representation of the flavonoid biosynthetic pathway was included as Fig. EV1, which makes the text easier to follow.

Q3 The cytoplasmic concentrations of the flavonoid glycosides I asked for were not measured for the technical reasons and these reasons were discussed. Instead the authors present a new experiments of comparison of the reaction of plants from different genotypes at the elevated concentrations of flavonoids. The data are convincing, clearly presented and described in Fig. EV4. Also the identification of flavonoids by LC-MS/MS is described in more detail in the material and methods section.

Q4 and Q5 Presentation of the data on Fig.2 is improved, as well as details of statistical analysis on Fig.4.

Q5 Additional data concerning the calcium spike and its relations to the chloroplast is presented on Fig.EV6, based on a new experiment.

I am satisfied with the improvements to the manuscript that the authors made upon my suggestion.

September 15, 2025

RE: Life Science Alliance Manuscript #LSA-2025-03328-TRR

Dr. Brenda S.J. Winkel
Virginia Tech
Biological Sciences
Fralin Life Sciences Institute
1015 Life Science Circle (MC0477)
Blacksburg, Virginia 24061

Dear Dr. Winkel,

Thank you for submitting your Research Article entitled "Antioxidant properties of dihydroxy B-ring flavonoids modulate circadian amplitude in Arabidopsis". It is a pleasure to let you know that your manuscript is now accepted for publication in Life Science Alliance. Congratulations on this interesting work.

DISTRIBUTION OF MATERIALS:

Again, congratulations on a very nice paper. I hope you found the review process to be constructive and are pleased with how the manuscript was handled editorially. We look forward to future exciting submissions from your lab.

Sincerely,
